# Euclid: Supercharging Multimodal LLMs with Synthetic High-Fidelity Visual Descriptions

## Abstract

Multimodal large language models (MLLMs) have made rapid progress in recent years, yet continue to struggle with low-level visual perception—particularly the ability to accurately describe the geometric details of an image. This capability is crucial for applications in areas such as robotics, medical image analysis, and manufacturing. To address this challenge, we first introduce *Geoperception*, a benchmark designed to evaluate an MLLM's ability to accurately transcribe 2D geometric information from an image. Using this benchmark, we demonstrate the limitations of leading MLLMs, and then conduct a comprehensive empirical study to explore strategies for improving their performance on geometric tasks. Our findings highlight the benefits of certain model architectures, training techniques, and data strategies, including the use of high-fidelity synthetic data and multi-stage training with a data curriculum. Notably, we find that a data curriculum enables models to learn challenging geometry understanding tasks which they fail to learn from scratch. Leveraging these insights, we develop *Euclid*, a family of models specifically optimized for strong low-level geometric perception. Although purely trained on synthetic multimodal data, Euclid shows strong generalization ability to novel geometry shapes. For instance, Euclid outperforms the best closed-source model, Gemini-1.5-Pro, by up to 54.52% on benchmark tasks.

## 1 Introduction

Multimodal large language models (MLLMs) have rapidly progressed in recent years, demonstrating remarkable potential in understanding and reasoning about the visual world through the powerful capabilities of large language models (LLMs) (Liu et al., 2024c;a; Achiam et al., 2023; Team et al., 2023; Hu et al., 2023; Tong et al., 2024a; Wang et al., 2024a). These models have showcased strong performance in tasks such as visual question answering (VQA) (Goyal et al., 2017), image captioning (Lin et al., 2014), and multimodal reasoning (Liu et al., 2023). As one recent example, LLaVA-NeXT-34B (Liu et al., 2024b) achieves an impressive 83.7% accuracy on the VQAv2 benchmark (Goyal et al., 2017), a comprehensive benchmark on natural image question answering.

While MLLMs achieve impressive results on tasks like VQA, their performance relies on high-level semantic extraction (Tong et al., 2024b); in contrast, they often fall short on *low-level visual perception*—i.e., the ability to accurately describe the geometric details of an image, such as the points, lines, angles, shapes, and spatial relationships among its constituent objects. This limitation becomes especially apparent in tasks requiring precise descriptions, such as mathematical visual problem solving (Zhang et al., 2024a; Lu et al., 2023), scientific visual understanding (Yue et al., 2024; Fu et al., 2024a), abstract visual reasoning (Jiang et al., 2024; Ahrabian et al., 2024), and even simple visual comprehension (Rahmanzadehgervi et al., 2024; Wang et al., 2024b). For example, when interpreting a graph diagram, precise recognition of edges is essential for extracting reliable information, and in geometry problem-solving, accurate identification of relationships between line segments and points is fundamental (Fu et al., 2024a). Beyond abstract tasks, precise visual perception is also vital in real-world applications, including spatial understanding for robotics, medical image analysis for accurate diagnosis, quality control in manufacturing to detect subtle defects, autonomous driving systems that rely on exact object localization or distance estimation, and augmented reality applications that demand precise overlay of virtual objects onto the real world.

In this paper, we aim to study the challenges of low-level visual perception in MLLMs, take steps to understand the root cause of their performance, and improve the models' capabilities in this area. We begin by developing a benchmark dataset specifically designed to evaluate precise geometric perception, which we call *Geoperception*. As a focused test bed, this benchmark targets 2D geometry tasks. Using this benchmark, we demonstrate the limitations of leading closed and open MLLMs, followed by a comprehensive empirical study to explore strategies for significantly improving their performance on geometric perception tasks. Our findings show the benefits of key factors such as model architecture, training techniques, and data strategies, including the use of synthetic data and multi-stage training with a data curriculum. Notably, we find that a data curriculum enables models to learn challenging low-level geometry understanding tasks, which they fail to learn from scratch, even when trained on a very large dataset. Using these lessons learned, we then train a family of models—using a carefully designed curriculum of synthetic data—that are specifically optimized for strong low-level geometric perception, which we call Euclid. We evaluate this family of models, and show that it excels on a variety of low-level geometric perception tasks.

Our main technical contributions are as follows:

- **Geoperception Benchmark:** We introduce a new benchmark dataset, *Geoperception*, derived from the Geometry-3K corpus (Lu et al., 2021), specifically designed to evaluate MLLMs' ability to accurately perceive surface-level geometric information without requiring complex inference or reasoning. Our benchmark reveals significant shortcomings in precise geometric perception across all leading visual-language models, both closed and open-source.

- **Empirical Study and Synthetic Data Engine:** To investigate the root cause of this performance, we conduct a detailed empirical exploration of MLLM architecture and training strategies. To aid in our investigation, we develop a synthetic data engine capable of generating high-fidelity visual representations of fundamental geometric elements. This study leads to key insights, such as the importance of certain architectural choices and the use of curriculum-based, multi-stage training with progressively more complex visual descriptions for improving low-level visual perception.

- **Euclid Model:** Leveraging the insights from our exploration and our synthetic data engine, we train Euclid, a series of multimodal LLMs tailored for high-quality geometric perception. Although purely trained on synthetic multimodal data and simple geometry shapes, Euclid achieves strong performance on the Geoperception benchmark, for instance, outperforming the best closed-source model, Gemini-1.5-Pro, by up to 54.52% on certain benchmark tasks.

## 2 BACKGROUND AND RELATED WORK

We provide an overview of prior efforts that assess and improve low-level perception and geometric reasoning in MLLMs, and highlight our contributions in data synthesis, evaluation, and training.

**Vision-Language MLLMs.** While recent iterations of LLMs feature a standardized model architecture and pretraining recipe, MLLMs still often differ in design choices for infusing visual inputs. One popular design is to align *continuous* visual features with the embedding space of a backbone LLM (Liu et al., 2024a;b; Dubey et al., 2024; McKinzie et al., 2024; Tong et al., 2024a; Beyer et al., 2024; AI, 2023; Wang et al., 2024a); another approach involves *tokenizing* visual inputs to be trained jointly with language tokens (Team et al., 2023; Team, 2024). These modules are often infused with a decoder-only LLM, but others have explored encoder-decoder architectures to integrate a more varied collection of modalities (Alayrac et al., 2022; Mizrahi et al., 2024; Ormazabal et al., 2024; Bachmann et al., 2024). Our study focuses on *decoder* MLLMs with a *continuous* visual encoder, and we carry out an empirical study to explore the effect of synthetic dataset mixture, training recipe, and encoder design (Liu et al., 2022; Radford et al., 2021; Zhai et al., 2023; Oquab et al., 2023).

**Geometry-Oriented MLLMs.** At the core of these choices is the hardness in designing a module adept in general visual reasoning (McKinzie et al., 2024; Tong et al., 2024a). In this work, we explore the optimal design of MLLMs specialized in low-level visual perception, a crucial aspect for (among other applications) multimodal mathematical understanding (Lu et al., 2023; Zhang et al., 2024a). This paper supplements prior efforts in improving mathematical reasoning (Gao et al., 2023; Zhang et al., 2024b; Zhuang et al., 2024; Li et al., 2024; Peng et al., 2024; Shi et al., 2024b) with a

detailed study on the effect of dataset mixture, curriculum, and visual encoder, to reach a recipe that elicits strong performance on geometric tasks (Kazemi et al., 2023) that require low-level perception.

**Evaluating Low-Level Perception.**    Many benchmarks (Rahmanzadehgervi et al., 2024) have reported that frontier-class MLLMs struggle with visual perception tasks, which are prerequisites for applications that emphasize low-level geometric perception (Chen et al., 2024; Fu et al., 2024b), including mathematical (Yue et al., 2024; Lu et al., 2023; Zhang et al., 2024a; Jiang et al., 2024) and spatial reasoning (Chen et al., 2024). These findings collectively identify that MLLMs exhibit a language prior (Lin et al., 2023)—a preference of textual inputs over visual inputs—leading to a performance gap between modalities (Wang et al., 2024b; Zhang et al., 2024a; Fu et al., 2024a). Meanwhile, there lacks a high-quality benchmark that evaluates low-level geometric perception in MLLMs, and the Geoperception benchmark represents a first effort to narrow this gap.

**Improving Low-Level Visual Perception.**    Many prior works study *data-driven* approaches to improve low-level perception skills. For example, Gao et al. (2023); Li et al. (2024); Zhuang et al. (2024) employ a standardized supervised finetuning recipe, and optionally adjust the training data mixture. This type of training data is often synthesized from text-only math problems (Lu et al., 2021; Trinh et al., 2024) or via rule-based systems (Kazemi et al., 2023). In parallel, Vishniakov et al. (2023); Shi et al. (2024a); Tong et al. (2024b) have explored the design space of visual encoders for general-purpose vision-language reasoning. We identify best practices over the union of these design spaces, and then train small MLLMs with strong performance in low-level perception tasks.

Lastly, several works (Schick et al., 2024; Surís et al., 2023; Hu et al., 2024) have opted to augment an MLLM with external APIs that process low-level features with specialized vision modules, such as object detection (Redmon et al., 2016), segmentation (Kirillov et al., 2023), and depth estimation (Yang et al., 2024b). While these agentic frameworks (Wu et al., 2023) present a promising alternative that directly addresses the shortcomings of visual encoders, they are limited by their scalability to novel use cases, and may be insufficient for precise tool routing that requires low-level perception as a primer (Picard et al., 2023; Wu et al., 2024; Buehler, 2024).

## 3  GEOPERCEPTION BENCHMARK

Recently, there has been a growing number of multimodal benchmarks across diverse domains beyond natural image understanding, including mathematical reasoning (Zhang et al., 2024a; Lu et al., 2023) and abstract visual reasoning (Jiang et al., 2024; Chia et al., 2024). Many of these prior works have realized the importance of accurate low-level visual perception. Specifically, Marvel (Jiang et al., 2024) introduces perception questions for various abstract reasoning patterns, and finds that the main bottleneck of MLLMs' performance on abstract visual reasoning is that they fail to accurately transcribe visual information into concepts; Mathverse (Zhang et al., 2024a) and IsoBench (Fu et al., 2024a) both test MLLMs on equivalent question represented by language and visual modalities, respectively. Both works find that language-only input always outperforms vision-language input, and that the vision component of MLLMs always fails to utilize low-level visual features. VDLM (Wang et al., 2024b) transcribes raster images into vector graphics and uses LLMs to reason over the SVG code. They find that although SVG code is not straightforward to understand, using LLMs to reason over SVG is consistently more effective than directly using MLLMs on original raster images. Blind-test (Rahmanzadehgervi et al., 2024) and BLINK (Fu et al., 2024b) also share similar findings with the works above.

**A Benchmark for Geometric Perception.** Although such shortcomings of MLLMs are commonly recognized, there is a lack of comprehensive benchmark that purely focuses on these abilities of MLLMs. Our goal is to construct a benchmark focusing solely on the perception ability of MLLMs, which is also representative enough of real-world applications. When humans perceive and memorize visual information, it is well-recognized that this procedure relies crucially on searching for the closest and simplest corresponding geometric shapes (Sablé-Meyer et al., 2022). We posit that geometric perception is a fundamental and broadly representative low-level visual perception ability in many applications. Hence, we select geometry understanding as our domain of dataset construction.

**Benchmark Tasks.** Over two thousand years ago, Euclid introduced five axioms that underpin all further geometric reasoning. These axioms involve establishing and extending lines using points

**PointLiesOnLine**

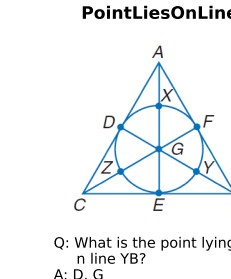

Q: What is the point lying o-
n line YB?
A: D, G

**PointLiesOnCircle**

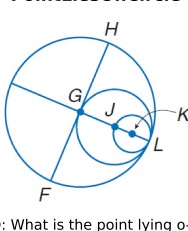

Q: What is the point lying o-
n circle with center G?
A: L, H, F

**AngleClassification**

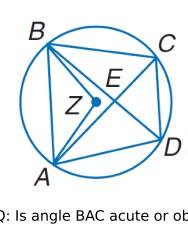

Q: Is angle BAC acute or obt-
use?
A: acute

**LineComparison**

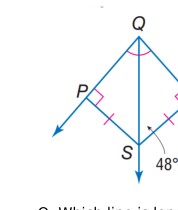

Q: Which line is longer, QS -
or SP?
A: QS

Figure 1: Four examples from our *Geoperception* dataset. The questions are sourced from the Geometry-3K corpus (Lu et al., 2021), which compiles problems from two widely-used high school textbooks. We perform filtering, validation, and generate question-and-answer text for each image.

(Axioms 1 and 2), constructing circles from a point and a radius (Axiom 3), and defining perpendicularity (Axiom 4) and parallelism (Axiom 5). Additionally, Euclid provided common notions regarding the properties of equality. To capture these aspects, we define five tasks in our Geoperception dataset: `PointLiesOnLine`, `PointLiesOnCircle`, `Parallel`, `Perpendicular` and `Equal`, and aditionally define `AngleClassification` and `LengthComparison` tasks to assess the model's understanding of angle and length measurements, resulting in a total of seven tasks. In geometric diagrams, perpendicularity, parallelism, and equality are often indicated by annotation symbols. Thus, we classify `Parallel`, `Perpendicular`, and `Equal` as annotated geometry understanding. Meanwhile, `PointLiesOnLine`, `PointLiesOnCircle`, `AngleClassification`, and `LengthComparison` fall under primitive geometry shape understanding, which includes both logical (`PointLiesOnLine`, `PointLiesOnCircle`) and numerical (`AngleClassification`, `LengthComparison`) tasks.

**Data Filtering.** Geoperception is sourced from the Geometry-3K (Lu et al., 2021) corpus, which offers precise logical forms for geometric diagrams, compiled from popular high-school textbooks. However, certain points in these logical forms are absent in the corresponding diagrams. To resolve this, we use GPT-4o-mini MLLM to confirm the presence of all points listed in the logical forms. This process filters the 3,002 diagrams to retain 1,584, where at least one logical form fully represents its points in the diagram. A random inspection of 100 annotations reveals only two errors, indicating high annotation accuracy.

Table 1: Statistics of the seven tasks in our Geoperception dataset, including the number of questions and images.

| Predicate | # Q | # I |
|---|---|---|
| PointLiesOnLine | 1901 | 924 |
| PointLiesOnCircle | 359 | 322 |
| Parallel | 106 | 101 |
| Perpendicular | 1266 | 456 |
| Equals | 4436 | 1202 |
| AngleClassification | 2193 | 1389 |
| LengthComparison | 1394 | 1394 |

**Converting Logical Forms Into Questions.** We convert logical forms into question-and-answer pairs for each of the seven tasks in Geoperception. In the `Equals` task, for example, we directly convert the logical form (e.g., `Equals(LengthOf(Line(Q, T)), 86)`) into a question-answer pair (e.g., `Q: What is the length of line QT as annotated? A: 86`). For `PointLiesOnLine`, two points on the line are chosen to form the question, with the remaining points on the line as the answer. Similarly, for `PointLiesOnCircle`, we ask which points lie on the circle, using its center as the basis for the question. For `Parallel` and `Perpendicular`, we represent each line by two points and query which other lines are parallel or perpendicular to it. In `AngleClassification`, we ensure the queried angle is in the range of $[10, 80] \cup [100, 170]$ degrees to avoid ambiguity. For `LengthComparison`, we ensure that the shorter line is less than 70% of the length of the longer line. Since multiple equivalent questions can be generated for a single logical form (e.g., a line containing five points generates $^5P_2$ equivalent questions), we randomly select one to avoid redundancy. Table 5 summarizes the question statistics for each task, as well as the number of images involved. Four examples from Geoperception are illustrated in Fig. 1

**Evaluation Details.** We evaluate seven leading MLLMs, both open source and closed source. The open source models include Molmo-7B-D (Deitke et al., 2024), Qwen2-VL-7B (Wang et al., 2024a), Llama-3.2-11B (Dubey et al., 2024), and Pixtral-12B (AI, 2023). The closed-source models include

GPT-4o-mini (Achiam et al., 2023), GPT-4o (Achiam et al., 2023), Claude-3.5-Sonnet (Anthropic, 2024), Gemini-1.5-flash (Team et al., 2023), and Gemini-1.5-pro (Team et al., 2023). Additionally, GPT-4o-mini without image input is used for generating the random baseline, employing the same textual instructions. To prevent stretching, all images are padded to square dimensions before being fed into the models. During evaluation of a given question by an MLLM, let $G$ denote the ground truth set of answers, and let $P$ denote the predicted set of answers; then the evaluation score is defined as

$$\text{Evaluation score} = \begin{cases} \dfrac{|P|}{|G|} & \text{if } P \subseteq G, \\ 0 & \text{otherwise.} \end{cases} \tag{1}$$

**Current MLLMs struggle to perceive low-level geometry annotations and relationships.** Despite the simplicity of Geoperception for humans, it remains a considerable challenge for even the most advanced commercial MLLMs. Notably, all models fall short of achieving 30% accuracy on the `PointLiesOnLine` task and do not outperform the text-only GPT-4o mini model in `AngleClassification` task. Closed source models generally outperform open source ones, with Gemini-1.5-pro attaining the highest overall score of 57.24%, followed by gemini-1.5-flash at 55.03%. Among open source models, Pixtral-12B achieves the best performance with an overall score of 41.84%. We show a comparison of all models on Geoperception in Table 2.

Certain models, such as GPT-4o-mini (Achiam et al., 2023) and Molmo-7B-D (Deitke et al., 2024), frequently either enumerate all potential components (e.g., all points in a diagram instead of the one on the lines) or every potential answer, leading to their poor accuracy scores.

Table 2: Performance (average evaluation score) of different models on Geoperception benchmark tasks. `POL`: PointLiesOnLine, `POC`: PointLiesOnCircle, `ALC`: AngleClassification, `LHC`: LineComparison, `PEP`: Perpendicular, `PRA`: Parallel, `EQL`: Equals. As the Random Baseline method, we use GPT-4o-mini, given the same textual instruction but without an image.

| Model | Logical | | Numerical | | Annotations | | | Overall |
|---|---|---|---|---|---|---|---|---|
| | POL | POC | ALC | LHC | PEP | PRA | EQL | |
| Random Baseline | 0.43 | 2.63 | 59.92 | 51.36 | 0.25 | 0.00 | 0.02 | 16.37 |
| *Open Source* | | | | | | | | |
| Molmo-7B-D (Deitke et al., 2024) | 1.75 | 35.73 | 56.77 | 16.79 | 1.10 | 0.00 | 0.81 | 16.14 |
| Llama-3.2-11B (Dubey et al., 2024) | 16.22 | 37.12 | 59.46 | 52.08 | 8.64 | 22.41 | 49.86 | 35.11 |
| Qwen2-VL-7B (Wang et al., 2024a) | 21.89 | 41.60 | 46.60 | 63.27 | 26.86 | 30.66 | 54.37 | 40.75 |
| Pixtral-12B (AI, 2023) | 22.85 | 53.21 | 47.33 | 51.43 | 22.53 | 37.11 | 58.45 | 41.84 |
| *Closed Source* | | | | | | | | |
| GPT-4o-mini (Achiam et al., 2023) | 1.65 | 61.19 | 48.84 | 69.51 | 10.04 | 4.25 | 44.75 | 34.32 |
| GPT-4o (Achiam et al., 2023) | 9.81 | 71.49 | 55.63 | 74.39 | 25.36 | 60.77 | 44.71 | 48.88 |
| Claude 3.5 Sonnet (Anthropic, 2024) | 25.44 | 68.34 | 42.95 | 70.73 | 22.00 | 64.39 | 66.36 | 51.46 |
| Gemini-1.5-Flash (Team et al., 2023) | 29.30 | 67.75 | 49.89 | 76.69 | 30.92 | 64.39 | 66.31 | 55.03 |
| Gemini-1.5-Pro (Team et al., 2023) | 24.42 | 69.80 | 57.96 | 79.05 | 39.60 | 77.59 | 52.27 | 57.24 |

## 4 EMPIRICAL STUDY ON MLLM DESIGN SPACE

We hypothesize that the lack of high-fidelity geometric visual perception data is one of the major reasons for the inability of today's MLLMs to effectively perceive basic geometric annotations and relationships. Although large-scale web-crawled image-text pairs cover a variety of domains, including geometry, the textual descriptions often lack the necessary specificity and depth. To address this issue, current studies in this domain (Gao et al., 2023; Shi et al., 2024b; Zhang et al., 2024b) typically construct a geometry or mathematical domain dataset and apply the same training strategy used for general-purpose MLLMs. For example, Math-LLaVA (Shi et al., 2024b) and multi-math (Peng et al., 2024) rely on GPT-4v or GPT-4o's vision ability to generate most of the question and answer pairs and image captions, which is essentially a form of model distillation. However, as evidenced by Table 2, GPT-4o and Gemini-1.5-Pro often struggle to answer certain types of questions, limiting the performance potential of resulting models. Furthermore, while works such as G-LLaVA (Gao et al., 2023), MAVIS (Zhang et al., 2024b), and Math-PUMA (Zhuang et al., 2024) utilize human

crafted logical forms or synthetic multimodal data to ensure the reliability of textual annotations, they often conflate low-level perception with problem-solving, and train models to directly solve multimodal geometry problems, without verifying if the model's low-level perception abilities are sufficient. As evidence, the best models in MAVIS (Zhang et al., 2024b) and Math-PUMA (Zhuang et al., 2024) evaluation results on Mathverse (Zhang et al., 2024a) still have a substantial gap of 26.8% and 28.7% between text-dominant versions and vision-only versions of problems[1], respectively. Furthermore, attempts to train MLLMs on low-level visual perception tasks (Wang et al., 2024b; Rahmanzadehgervi et al., 2024) have also struggled to achieve satisfactory in-domain performance or generalize effectively. In this section, we aim to address these challenges.

In recent work, the design space for MLLMs has been closely explored (McKinzie et al., 2024; Tong et al., 2024a; Shi et al., 2024a). However, most studies rely on general multimodal benchmarks to evaluate design efficacy, which often do not effectively assess visual understanding capabilities (Tong et al., 2024a), thereby limiting their utility in evaluating precise visual perception. Additionally, our findings indicate that, under the current multimodal instruction tuning paradigm, MLLMs exhibit significant challenges in performing zero-shot basic visual perception tasks. Therefore, we revisit the design space of MLLMs and employ task-specific tuning to investigate the potential of diverse multimodal designs.

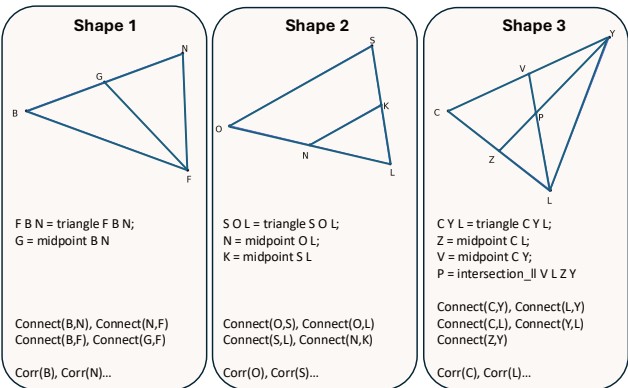

Figure 2: Three **synthetic geometry shapes** used for training, and their corresponding high-fidelity visual descriptions, which include the relationships between each geometry component, the existence of line connections, and numerical attributes such as point coordinates. Our dataset generation engine generates question and answer pairs based on these visual descriptions. The specific process of question answer pair generation is detailed in Appendix C.

**Geometry Shape Generation Engine.** Unlike natural images, geometric images can be generated programmatically, enabling the creation of nearly infinite numerical instances for each conceptual shape. Our geometry shape generation engine builds on Alphageometry (Trinh et al., 2024) due to the superior performance of the language model trained on the dataset generated by this engine. Specifically, we introduce three visualization enhancements: (1) an additional input to control the connections between points, (2) increased randomness in deriving numerical instances from conceptual shapes, and (3) adjustments to the canvas range to ensure visibility of all geometry components.

Table 3: Summary of Visual Encoders

| Model | Params | Objective |
|---|---|---|
| ConvNeXt Large@512 | 200M | CLIP |
| ConvNeXt XXLarge@512 | 847M | CLIP |
| ViT-g/14@224 | 1.01B | CLIP |
| ViT-H/14@224 | 632M | CLIP |
| ViT-L/14@336 | 304M | CLIP |
| ViT-L/14@224 | 303M | CLIP |
| SigLIP@384 (ViT) | 428M | CLIP-like |
| SigLIP@224 (ViT) | 428M | CLIP-like |
| DINOv2 Giant@224 (ViT) | 1.14B | Self-Sup |
| DINOv2 Large@224 (ViT) | 304M | Self-Sup |

**Exploration Overview.** We study the choice of visual encoder architecture, the choice between tuning or freezing the encoder, and different data composition/training strategies. For visual encoders, we investigate two families of architectures: Vision Transformer (ViT) (Dosovitskiy, 2020) and ConvNeXT (Liu et al., 2022); as well as two visual representation learning objectives: language-supervised learning (Radford et al., 2021) and self-supervised learning (Oquab et al., 2023). Additionally, we examine the impact of varying encoder sizes and the number of visual tokens. The list of visual encoders and their parameters are shown in Table 3. For LLMs, we use Qwen-2-1.5B-instruct (Yang et al., 2024a). For the multimodal connection, we use a two layer MLP as multimodal encoder following LLaVA-1.5 (Liu et al., 2024a). We leave exploring visual connector choices and scaling the size of LLMs as future work.

---

[1]In Mathverse, text-dominant is the version where the problem is mainly represented by text, while in the vision-only version an equivalent problem is represented purely by image.

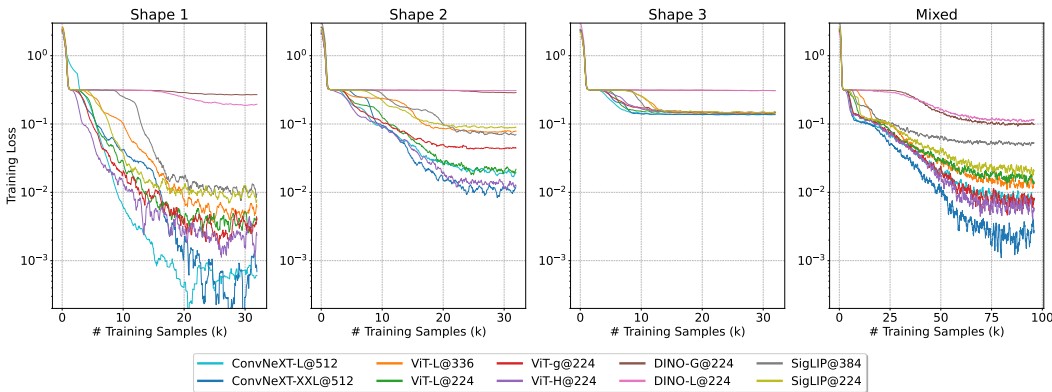

Figure 3: Training loss curve comparing ten visual encoders, with a fixed multimodal encoder and LLM. Training losses are window-smoothed using a window size of 10 for better visibility. Losses are log-scaled to demonstrate their difference in smaller values.

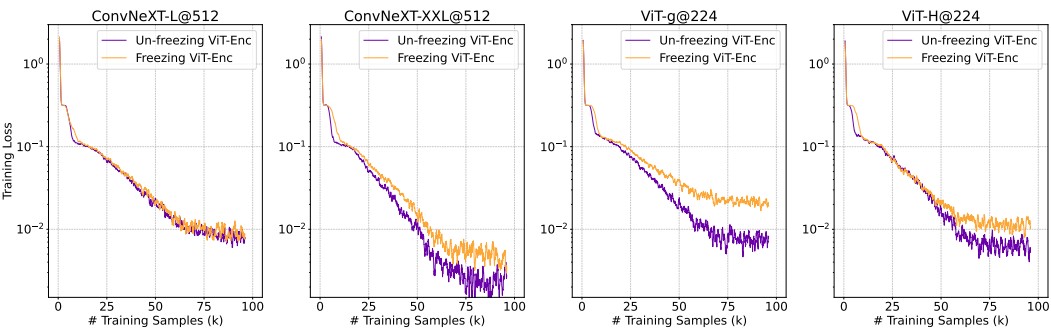

Figure 4: Training loss curve comparing freezing versus unfreezing the visual encoder. This is shown on the four best-performing architectures. Loss curves are window-smoothed with a window size of 10 for better visibility. Losses are log-scaled to demonstrate their difference in smaller values.

### 4.1 RESULTS OF EMPIRICAL STUDY AND LESSONS

We use learning efficacy as the measure for evaluating visual encoders. The task chosen for this exploration is `PointLiesOnLine`, the most fundamental task in Geoperception. In `PointLiesOnLine` questions, each line must have at least three points to form a valid query. To support this evaluation, we designed three basic geometric conceptual shapes of increasing complexity, containing 1, 2, and 4 valid lines respectively. These shapes are illustrated in Fig. 2. We separately train our models on three shapes, each shape for 500 steps with a batch size of 64. In addition, we mix together data of the three shapes and train our models on 1,500 steps, as our fourth experiment. We now present the three main lessons that we determined via our empirical study.

**Lesson 1: CNN architecture performs better than ViT.** We actively tune all of the parameters in the MLLM, including the visual encoder, and show the training loss curve of ten different visual encoders in Fig. 3. We find that ConvNeXt-XXLarge consistently learns the geometric concept the fastest among all of the visual encoders. Moreover, although with only 200M parameters, ConvNeXT-Large shows competitive learning performance with the vision transformers which are 3-5 times larger. Self-supervised learning (SSL) visual encoders, DINO-v2, struggles to learn the geometry concept; we hypothesis this is due to the weak vision-language representation in these models. Surprisingly, although the SigLIP-family is widely-recognized as a better visual encoder (Tong et al., 2024a), we find that their performance in learning basic visual geometry attributes is limited.

In addition, image resolution does not make a significant role on such potential. Specifically, CLIP-L@336 and SigLIP@384, higher-resolution visual encoders, learn the task consistently slower than CLIP-L@224 and SigLIP@224, respectively. Moving forward, our analysis will focus on four top-performing visual encoders: ConvNeXt-Large, ConvNeXt-XXLarge, ViT-g and ViT-H.

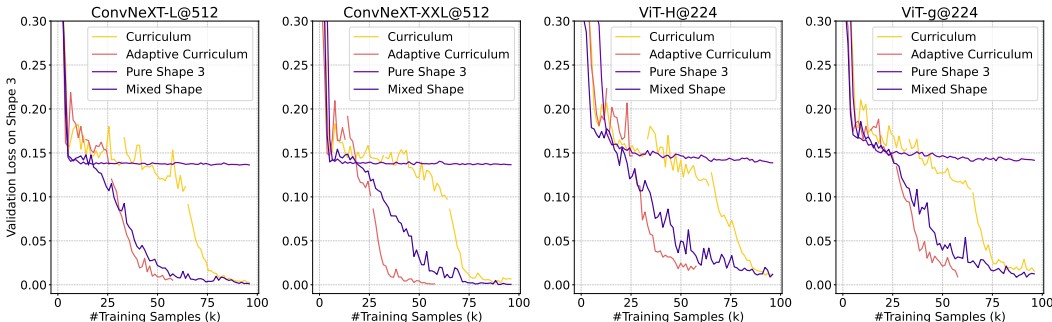

Figure 6: Comparison between four training strategies with the objective of effectively learning a task on a complex shape. For all curves we show the validation loss on the same held-out dataset (comprising samples of shape 3).

**Lesson 2: Tuning the visual encoder is beneficial.** We next study the effect of tuning versus freezing the visual encoder. In Fig. 4, we show the loss curves of tuning and freezing visual encoders. We find that tuning the visual encoder consistently helps the model learn low-level geometry relationships faster and better, in comparison with using a frozen encoder.

**Lesson 3: Curriculum learning unleashes full potential.** Finally, we study training data composition. In Fig. 3, we observe that all models fail to converge on *Shape 3* (the most challenging shape in our experimental setup with four valid query lines). However, when using a mixed training set of all three shapes, some visual encoders achieve convergence, despite using the same amount of data for *Shape 3*. We hypothesize that including simpler shapes (*Shape 1* and *Shape 2*) aids the model in learning more complex shapes (*Shape 3*). To test this hypothesis, we report the loss functions for *Shapes 1*, *2*, and *3* separately during the mixed training of ConvNeXt-XXLarge, in Fig. 5. We notice a plateau in the loss curve for *Shape 3* until the model has trained on approximately 20K samples. During this period, the losses for *Shape 1* and *Shape 2* continue to decrease. This suggests that learning easier shapes can significantly help the model tackle more challenging shapes, comparing with directly learning the challenging ones, this finding align with the principles of curriculum learning.

While mixed training enables effective spontaneous curriculum learning, we investigate whether a structured curriculum can further enhance model efficiency on challenging shapes. To this end, we train the model sequentially from simple to more complex shapes and compare the loss on a separate validation set of *Shape 3*. To avoid forgetting, we apply smoothed data at each stage: 80% from the current shape and 10% from each of the others. We refer to this as a staged curriculum strategy. The results are shown in Fig. 6. We find that all of the models fail to converge when trained purely on *Shape 3*, In contrast, the staged curriculum strategy, shown by the yellow curve, consistently reaches a good validation loss on *Shape 3* after training. To further optimize its

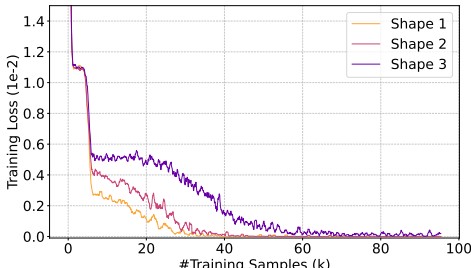

Figure 5: The breakdown of training loss curve of shape 1,2 and 3 during the mixed training of three shapes.

efficiency, we reduce the training data for the two simpler shapes to 40% of their original volume. This approach, represented by the orange line in Fig. 6, proves more efficient than mixed training. For even greater efficiency, we propose adapting the dataset on-the-fly based on monitoring the loss during training on generated images, as described in Section 5.

## 5 EUCLID

In this section, we take all of the lessons we learned in the previous sections and train Euclid, a family of MLLMs specifically designed for strong low-level geometric perception.

**On-the-fly progressive training.** Drawing inspiration from the effectiveness of curriculum learning, instead of constructing a static training data, we introduce an adaptive dataset generation method-

ology with our dataset engine, where we monitor the model's performance and dynamically adjust the distribution of training data (i.e., the curriculum stage) based on this performance. Specifically, for a certain task, we create $N$ stages of training dataset shapes with progressively increasing geometric complexity. During training, the model starts by training on the first (simplest) dataset stage. The model is evaluated when it finishes a training round, using a held-out validation set from the same distribution as the current dataset stage. Upon evaluation, if the model achieves an accuracy exceeding a predefined threshold $\theta$, the framework advances the task to the next stage. Formally, the update rule for advancing stages is given by:

$$\text{if accuracy}_s > \theta \quad \Rightarrow \quad c \leftarrow c + 1. \tag{2}$$

The model is trained on a total of $M$ rounds and $K$ steps within each round. Similar to Section 4, we smooth our dataset distribution over all stages using an exponential attenuation function:

$$\text{ratio}_s = \exp\left(-\alpha \cdot |\text{stage}_s - c|\right), \tag{3}$$

where $\alpha$ denotes the attenuation rate. Eq. (3) ensures that stages proximal to the current stage receive higher sampling probabilities.

**Specifications.** For models, we select the best visual encoder architecture we found in our investigation, ConvNeXt, including ConvNeXt-Large@512 and ConvNeXt-XXLarge@512, and keep the same multimodal connector (2 layers MLP) and LLM (Qwen2-1.5B-instruct). For tasks, we focus on four primitive tasks from the Geoperception benchmark which are easily scalable using our dataset generation engine : `PointLiesOnLine`, `PointLiesOnCircle`, `AngleClassification`, and `LengthComparison`. We both separately and jointly train our model on each of the tasks, and test our resulting model on the corresponding tasks in Geoperception that the model is trained on. The accuracy threshold for advancing training stage $\theta$ is set to $0.99$. All models are trained on $N = 3$ stages with manually curated geometry shapes and $M = 6$ rounds with $K = 500$ steps in each round, and the batch size is $64$ for each training step.

Table 4: Performance comparison between Euclid and the best leading open source and closed source MLLMs on the four tasks: POL, POC, ALC, LHC. Note that Euclid is *not* trained on any of the in-distribution data from the benchmark tasks below. We report the performance of both the separately trained model and the jointly trained models.

| Model | POL | POC | ALC | LHC | Average |
|---|---|---|---|---|---|
| Pixtral-12B (AI, 2023) | 22.85 | 53.21 | 47.33 | 51.43 | 43.71 |
| Gemini-1.5-Pro (Team et al., 2023) | 24.42 | 69.80 | 57.96 | 79.05 | 57.81 |
| Euclid-ConvNeXt-Large@512 | 77.17 | 73.06 | 61.06 | 77.12 | 72.10 |
| Euclid-all in one-ConvNeXt-Large@512 | 59.52 | 66.18 | 71.41 | 74.96 | 68.02 |
| Euclid-ConvNeXt-XXLarge@512 | 78.94 | 67.94 | 61.51 | 78.19 | 71.65 |
| Euclid-all in one-ConvNeXt-XXLarge@512 | 55.22 | 70.65 | 66.85 | 74.03 | 66.69 |

**Evaluation results.** The results are shown in Table 4. While Euclid is trained on simple synthetic geometric shapes and uses only a 1.5B language model, demonstrates superior performance on average across four primitive tasks compared to existing leading MLLMs, exhibiting strong generalization to real-world geometric shapes. Notably, in the `PointLiesOnLine` task, which is particularly challenging for existing MLLMs, Euclid achieves up to 78.94% accuracy, nearly three times the performance of Gemini-1.5-Pro. On `LengthComparison` tasks, Euclid's performance is slightly outclassed by Gemini-1.5-pro, on other tasks, Euclid keeps higher or similar performance with the leading MLLMs. Interestingly, when models are jointly trained on multiple tasks, certain tasks, such as `PointLiesOnLine`, show slightly reduced performance compared with only training on the given task, which contrasts with the expected benefits of multi-task training (Liu et al., 2024a; Wei et al., 2021). We hypothesize two main reasons for this phenomenon. First, in general multimodal instruction tuning, datasets are often limited or insufficient, and training on multiple tasks can compensate for this by expanding

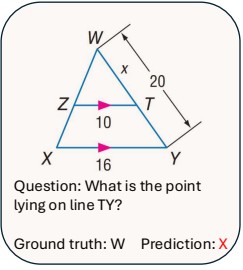

Question: What is the point lying on line TY?

Ground truth: W    Prediction: X

Figure 7: An error case where Euclid fails to predict the correct point on a line, potentially distracted by the annotation "x".

the data volume. However, our dataset generation engine can produce infinite samples for exhaustive task-specific training, thus diminishing the advantage of multi-task learning. Second, the decrease in performance may be related to the limitations of our 1.5B language model. For example, many `PointLiesOnLine` questions in Geoperception involve circles, which could lead the model to confuse these with `PointLiesOnCircle`.

**Error analysis.** We take a deep look into Euclid's prediction on Geoperception, we find that our model's performance is hindered when diagrams are heavily annotated. An example is shown in Fig. 7, where a line is annotated by "x", confusing the model from choosing the correct point. Incorporating training data that distinguish different diagram annotation types could potentially help the model with such scenarios.

## 6 CONCLUSION AND FUTURE WORK

In this work, we highlight the importance of accurate low-level visual perception in MLLMs. To this end, we first introduce Geoperception, a large-scale multimodal benchmark focused exclusively on geometry-domain visual perception. We evaluate leading MLLMs on Geoperception, find that even top models such as Gemini-1.5-Pro struggle significantly it, although it is straightforward for humans. We then conduct an empirical study to explore the design space of MLLM training and architectures using the dataset generated by a geometric high-fidelity synthetic-data engine that we develop. Our study indicate that convolutional neural network visual encoders outperform vision transformers in our tasks; tuning the visual encoder generally enhances performance; and employing a curriculum-based training approach yields much more model potential than direct task training. Based on insights from this study, we develop Euclid, a model trained purely on high-fidelity synthetic generated data, which generalizes effectively to real-world geometric shape understanding tasks, surpassing the leading MLLMs by a substantial margin.

**Future work.** Our work examines the potential of using synthetic multimodal data to improve MLLM performance in low-level geometric perception tasks. However, there are still directions that remain under-explored: (1) Using a more-diverse training dataset. Currently, the text portion of our synthetic multimodal training data uses a restricted set of templates, and the model trained on such templates could fail to generalize to other question types; it could therefore be beneficial to increase the diversity of our instruction-following formats. (2) Automatic curriculum learning. Incorporating a more diverse dataset, including varied geometric shapes and different domain dataset, introduces challenges in defining the learning order. Rule based definition and manual curation may become impractical, necessitating automated strategies like hard negative sampling to organize the curriculum based on training loss or testing accuracy. This approach could streamline the process, reduce human effort, provide more suitable and efficient curriculum learning orders. (3) Generalizing to other task domains. In this work, our study is focused on data from 2D geometry, as it provides a focused test bed of fundamental tasks. We believe the lessons we learn from this domain can be effectively generalized to a broader set of downstream domains that benefit from high-quality low-level visual perception.

## REPRODUCIBILITY STATEMENT

In Section 3, we provide a comprehensive description of the procedure for generating the Geoperception benchmark. This includes employing GPT-4o-mini for dataset filtering and detailing the conversion of logical forms into questions and answers. Evaluation prompts for MLLMs on different types of Geoperception questions are presented in Appendix B. For model architecture exploration, we specify the visual encoders and provide corresponding Hugging Face links in Table 3. Additionally, we outline the LLMs and multimodal connector architectures used. For our Euclid model, we include all geometry shape code used for training, along with demonstration diagrams and pseudocode for generating training questions and answers.

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

# APPENDIX

## A GEOPERCEPTION BENCHMARK DETAILS

In Table 5, we provide more details on the Geoperception benchmark, such as the number of logic forms present before and after filtering, the number of questions, and the number of images. `AngleClassification` and `LineComparison` are directly derived from points coordinates without filtering.

| Predicate | # LF Before Filter | # LF After Filter | # Q | # I |
|---|---|---|---|---|
| PointLiesOnLine | 6988 | 2567 | 1901 | 924 |
| PointLiesOnCircle | 1966 | 1240 | 359 | 322 |
| Parallel | 222 | 123 | 106 | 101 |
| Perpendicular | 1111 | 680 | 1266 | 456 |
| Equals | 6434 | 4123 | 4436 | 1202 |

Table 5: Statistics of the five predicates in our Geoperception dataset. Including number of logic forms before filter, after filter and the number of questions and images.

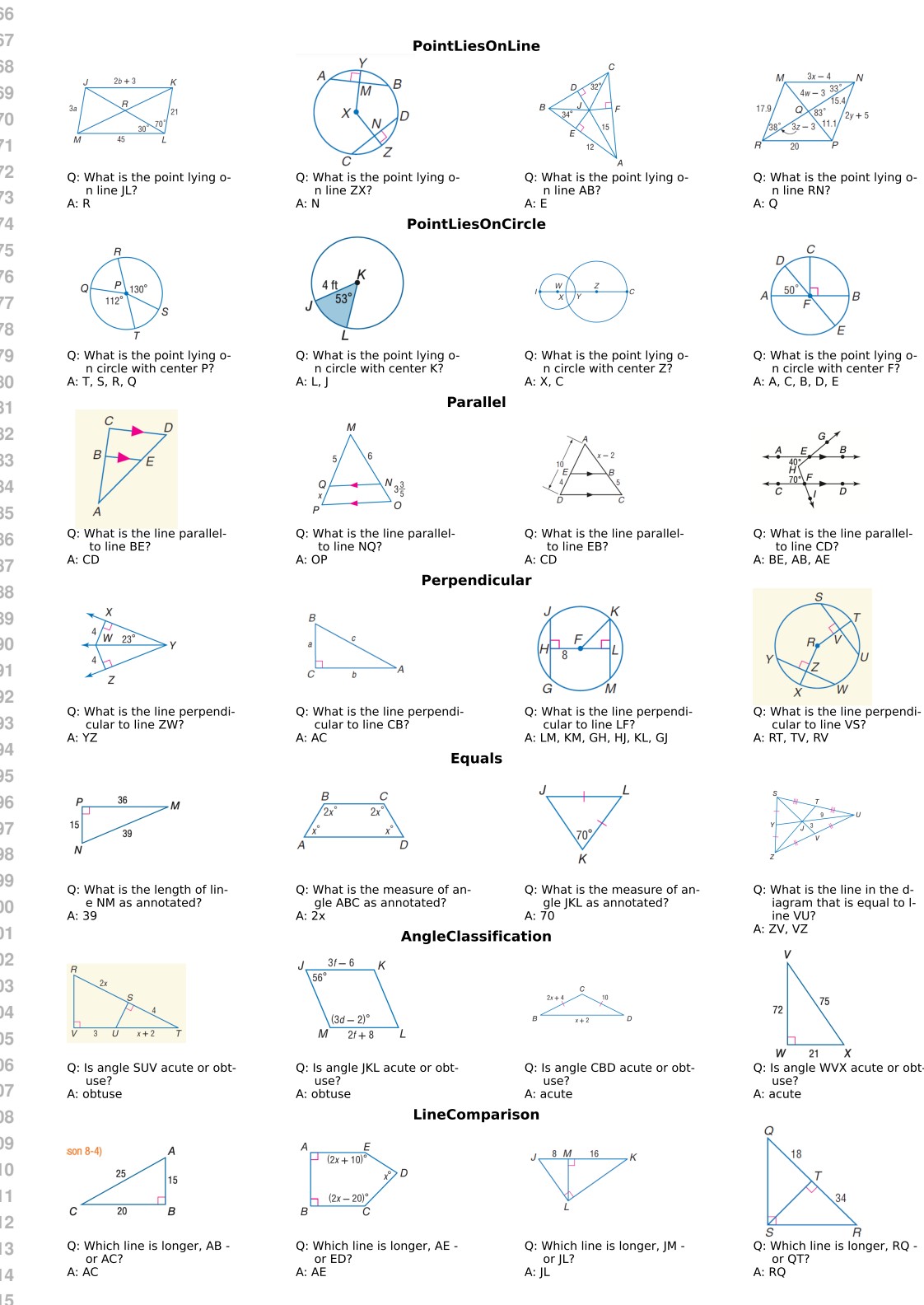

Figure 8: Examples of our *Geoperception* dataset.

# B  PROMPTS FOR THE GEOPERCEPTION DATASET EVALUATION

---

**PROMPT TEMPLATE FOR THE POINTLIESONLINE TASK**

```
Answer me directly just with the all points lie on the line
mentioned in the question (do not include the point mentioned in
the question).
Answer template:
  (If only one point) The other point is:  "your point".
Or
  (if multiple points) The other points are:  "your points".
For example:
  The other point is:  A
Or
  The other points are:  A, B, C
```

Figure 9: TEMPLATE FOR THE POINTLIESONLINE TASKS

---

**PROMPT TEMPLATE FOR THE POINTLIESONCIRCLE TASK**

```
Answer me directly just with the all points lie on the circle
mentioned in the question.
Answer template:
  (If only one point) The point is:  "your point".
Or
  (If multiple points) The points are:  "your points".
For example:
  The point is:  A
Or:
  The points are:  A, B, C
```

Figure 10: TEMPLATE FOR THE POINTLIESONCIRCLE TASKS

---

**PROMPT TEMPLATE FOR THE PARALLEL TASK**

```
Answer me directly just with the all lines which are parallel
to the line mentioned in the question (do not include the line
mentioned in the question).
Answer template:
  (If only one line) The line is:  "your line".
Or
  (If multiple lines) The lines are:  "your lines".
For example:
  The line is:  BC
Or:
  The lines are:  BC, DE
```

Figure 11: TEMPLATE FOR THE PARALLEL TASKS

PROMPT TEMPLATE FOR THE PERPENDICULAR TASK

```
Answer me directly just with the all lines which are perpendicular
to the line mentioned in the question (do not include the line
mentioned in the question).
Answer template:
  (If only one line) The line is:  "your line".
Or
  (If multiple lines) The lines are:  "your lines".
For example:
  The line is:  BC
Or:
  The lines are:  BC, DE
```

Figure 12: TEMPLATE FOR THE PERPENDICULAR TASKS

PROMPT TEMPLATE FOR THE EQUALS TASK

```
Answer me directly just with the annotations presented on the
image.
Answer template:
  The annotation is:  "your annotation".
For example:
  The annotation is:  2x+4
Or:
  The annotations is:  90
```

Figure 13: TEMPLATE FOR THE EQUALS TASKS

PROMPT TEMPLATE FOR THE ANGLE CLASSIFICATION TASK

```
Answer me directly just with the classification of the angle
mentioned in the question.
Answer template:
  The angle is:  "your angle".
For example:
  The angle is:  acute
Or:
  The angle is:  obtuse
```

Figure 14: TEMPLATE FOR THE ANGLE CLASSIFICATION TASKS

PROMPT TEMPLATE FOR THE LENGTH COMPARISON TASK

```
Answer me directly just with the longer line mentioned in the
question.
Answer template:
  The longer line is:  "your line".
For example:
  The longer line is:  BC
Or:
  The longer line is:  DE
```

Figure 15: TEMPLATE FOR THE LENGTH COMPARISON TASKS

# C DETAILS FOR TRAINING DATA ENGINE

In this section, we provide all geometry shapes we use for Euclid training, including the pseudocode for generating text describing the geometry shapes and diagram examples.

## C.1 PSEUDOCODE FOR TRAINING TEXTUAL DATASET SYNTHESIS

---

**Algorithm 1** Data Synthesis for the POINTLIESONLINE Task

---

```
1: Input: data_info, points_set
2: Output: data
3: for points_set ∈ data_info do
4:     for (A, B) ∈ permutations(points_set, 2) do
5:         all_rest_points ← [p for p in points_set if p not in [A, B]]
6:         for rest_points ∈ permutations(all_rest_points) do
7:             verb_agreement ← 'is' if len(rest_points) == 1 else 'are'
8:             rest_points ← [f"{p}" for p in rest_points]
9:             rest_points ← sorted(rest_points)
10:            question ← 'What is the point lying on line ' + A + B + '?'
11:            answer ← 'The point lying on line ' + A + B + ' ' + verb_agreement + ' ' + ', '.join(rest_points)
12:            gt ← ''.join(rest_points)
13:            data ← {'question': question, 'answer': answer, 'gt': gt}
14:        end for
15:    end for
16: end for
```

---

---

**Algorithm 2** Data Synthesis for the POINTLIESONCIRCLE Task

---

```
1: Input: data_info
2: Output: data
3: point_set ← random.choice(list(data_info.items()))
4: center_point ← point_set[0]
5: target_points ← point_set[1]
6: target_points ← sorted(target_points)
7: question ← 'What are the point lying on circle ' + center_point + '?'
8: answer ← 'The point lying on circle ' + center_point + ' are ' + ', '.join(target_points)
9: gt ← ''.join(target_points)
10: data ← {'question': question, 'answer': answer, 'gt': gt}
```

---

---

**Algorithm 3** Data Synthesis for the ANGLECLASSIFICATION Task

---

1: **Input:** `data_info`
2: **Output:** `data`
3: `angle ← data_info`
4: `angle_options ← [f'{angle[1][0]}{angle[1][1]}{angle[1][2]}',`
   `f'{angle[1][2]}{angle[1][1]}{angle[1][0]}']`
5: `angle_letter ← random.choice(angle_options)`
6: `angle_class ← 'acute' if angle[0] < 90 else 'obtuse'`
7: `question ← 'Is angle ' + angle_letter + ' acute or obtuse?'`
8: `answer ← 'Angle ' + angle_letter + ' is ' + angle_class`
9: `gt ← angle_class`
10: `data ← {'question': question, 'answer': answer, 'gt': gt}`

---

**Algorithm 4** Data Synthesis for the LINECOMPARISON Task

---

1: **Input:** `data_info`
2: **Output:** `data`
3: `names ← [data_info[0][1], data_info[1][1]]`
4: `lengths ← [data_info[0][0], data_info[1][0]]`
5: **if** `lengths[0] > lengths[1]` **then**
6:    `longer_name, shorter_name ← names[0], names[1]`
7: **else**
8:    `longer_name, shorter_name ← names[1], names[0]`
9: **end if**
10: `data ← [`
11:   `{ 'question': 'Which line is longer, ' + longer_name + ' or '`
    `+ shorter_name + '?',`
12:    `'answer': 'The longer line is ' + longer_name,`
13:    `'gt': longer_name },`
14:    `{ 'question': 'Which line is longer, ' + shorter_name + ' or`
   `' + longer_name + '?',`
15:    `'answer': 'The longer line is ' + longer_name,`
16:    `'gt': longer_name }`
17: `]`

---

## C.2  GEOMETRY SHAPES USED FOR EUCLID TRAINING

---

**GEOMETRY SHAPE GENERATION CODE**

```
PointLiesOnLine:
  (stage 1) A B C = triangle A B C; D = midpoint B C
  (stage 2) A B C = triangle A B C; D = midpoint A B; E = midpoint A C
  (stage 3) A B C = triangle A B C; D = midpoint B C; E = midpoint A C; F =
intersection_ll A D B E
PointLiesOnCircle:
  (stage 1) A B = segment A B; C = on_circle C A B
  (stage 1) A B = segment A B; C = on_circle C A B
  (stage 1) A B = segment A B; C = on_circle C A B; D = on_circle D A B
  (stage 1) A B = segment A B; C = on_circle C A B; D = on_circle D A B; E = on_circle E
A B
  (stage 1) A B = segment A B; C = on_circle C A B; D = on_circle D A B; E = on_circle E
A B; F = on_circle F A B
  (stage 1) A B = segment A B; C = on_circle C A B; D = on_circle D A B; E = on_circle E
A B; F = on_circle F A B; G = on_circle G A B
  (stage 2) A B = segment A B; C = on_circle C A B; D = midpoint A B
  (stage 2) A B = segment A B; C = on_circle C A B; D = midpoint A B
  (stage 2) A B = segment A B; C = on_circle C A B; D = midpoint A B; E = on_circle E A
B
  (stage 2) A B = segment A B; C = on_circle C A B; D = midpoint A B; E = on_circle E A
B; F = on_circle F A B
  (stage 2) A B = segment A B; C = on_circle C A B; D = midpoint A B; E = on_circle E A
B; F = on_circle F A B; G = on_circle G A B
  (stage 2) A B = segment A B; C = on_circle C A B; D = midpoint A B; E = on_circle E A
B; F = on_circle F A B; G = on_circle G A B; H = on_circle H A B
  (stage 3) A B = segment A B; C = on_circle C A B; D = midpoint A B; E = midpoint A C
  (stage 3) A B = segment A B; C = on_circle C A B; D = midpoint A B; E = midpoint A C;
F = on_circle F A B
  (stage 3) A B = segment A B; C = on_circle C A B; D = midpoint A B; E = midpoint A C;
F = on_circle F A B; G = on_circle G A B
  (stage 3) A B = segment A B; C = on_circle C A B; D = midpoint A B; E = midpoint A C;
F = on_circle F A B; G = on_circle G A B; H = on_circle H A B
  (stage 3) A B = segment A B; C = on_circle C A B; D = midpoint A B; E = midpoint A C;
F = on_circle F A B; G = on_circle G A B; H = on_circle H A B; I = on_circle I A B
  (stage 3) A B = segment A B; C = on_circle C A B; D = midpoint A B; E = on_circle E A
B; F = on_circle F A B; G = on_circle G A B; H = on_circle H A B; I = midpoint B C
  (stage 3) A B = segment A B; C = on_circle C A B; D = midpoint A B; E = midpoint B C
  (stage 3) A B = segment A B; C = on_circle C A B; D = midpoint A B; E = lc_tangent E C
A
  (stage 3) A B = segment A B; C = on_circle C A B; D = midpoint A B; E = on_circle E A
B; F = on_circle F A B; G = on_circle G A B; H = lc_tangent H C A
AngleClassification:
  (stage 1) A B C = triangle A B C
  (stage 2) A B = segment A B; C D = segment C D
  (stage 3) A B C = triangle A B C
  (stage 3) A B C = triangle A B C; D = midpoint B C
LineComparison:
  (stage 1) A B C = triangle A B C
  (stage 1) A B C = triangle A B C
  (stage 1) A B C = triangle A B C
  (stage 2) A B C = triangle A B C; D = midpoint B C
  (stage 3) A B C = triangle A B C; D = midpoint A B; E = midpoint A C
```

Figure 16: GEOMETRY SHAPE GENERATION CODE FOR EUCLID TRAINING

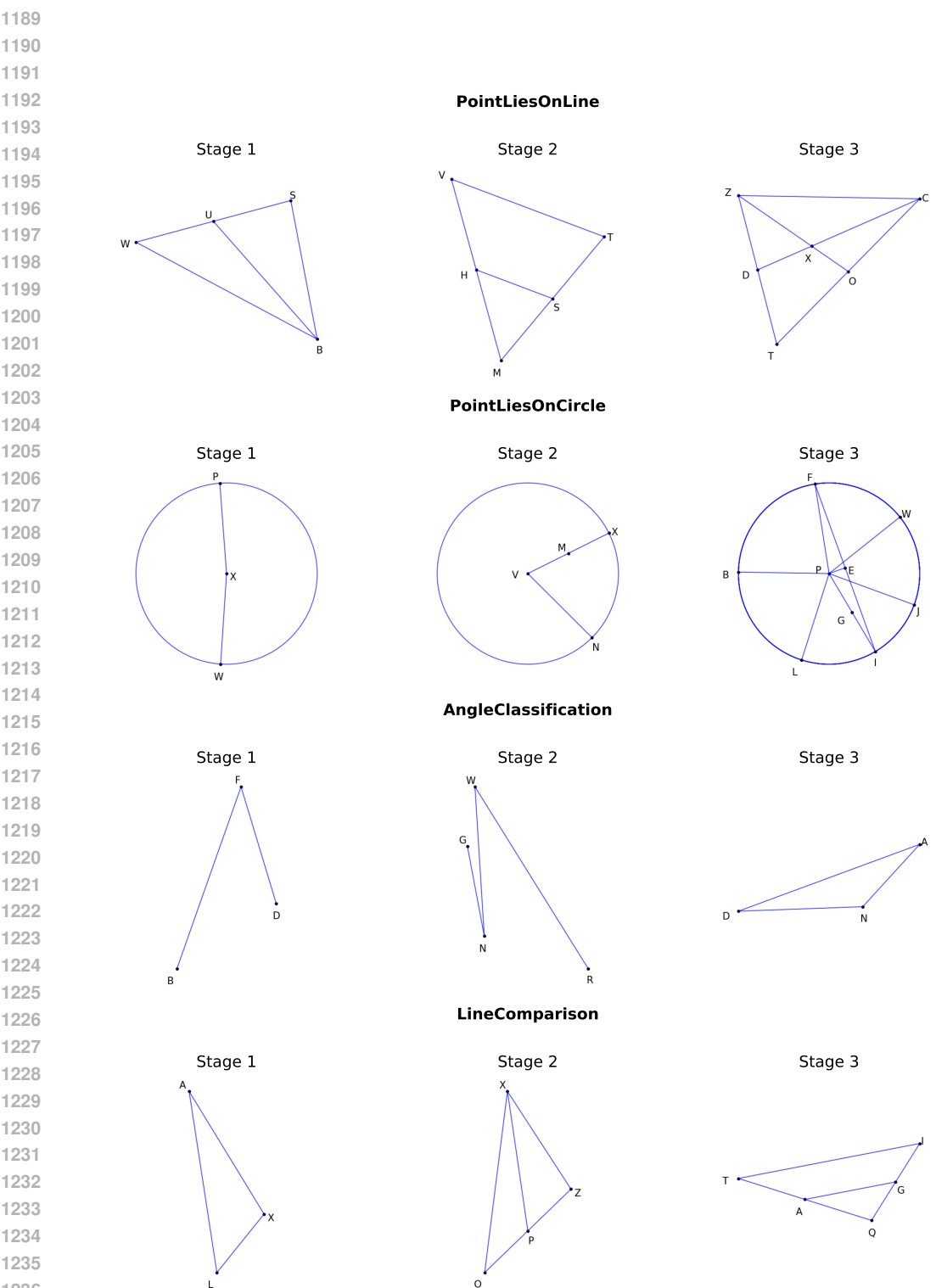

Figure 17: Examples of the geometry diagrams used to train Euclid, the diagrams are generated by our dataset engine.

# D    ADDITIONAL RESULT FIGURES IN REBUTTAL PHASE

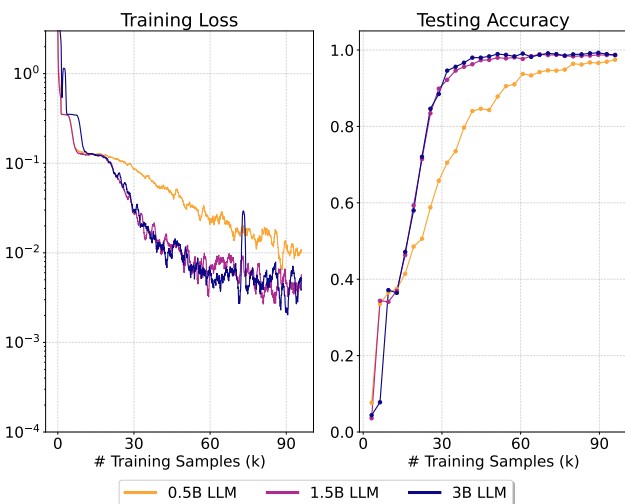

Figure 18: Training loss and testing accuracy curve comparing three choices of LLM size with a fixed visual encoder and multimodal connector. Training losses are window-smoothed using a window size of 10 for better visibility.

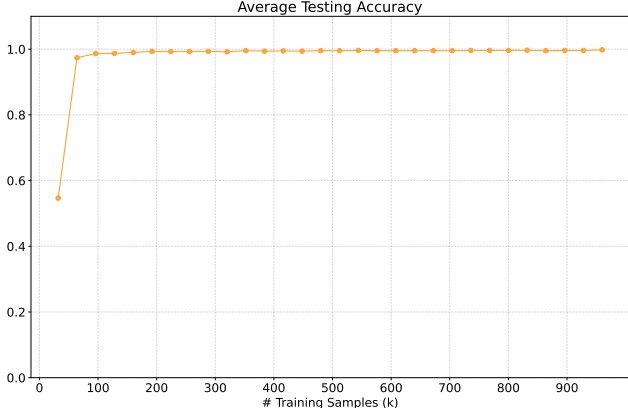

Figure 19: The test accuracy curve when expanding our training dataset volume to 1 million dataset, the model and dataset setting is the same as the last sub-figure in Fig. 3.

Table 6: Performance (average evaluation score) of different models on Geoperception benchmark tasks. The evaluation score is computed as a binary indicator where Evaluation score = 1 if predictions ($P$) are a subset of ground truth ($G$), and Evaluation score = 0 otherwise. POL: PointLiesOnLine, POC: PointLiesOnCircle, ALC: AngleClassification, LHC: LineComparison, PEP: Perpendicular, PRA: Parallel, EQL: Equals. As the Random Baseline method, we use GPT-4o-mini, given the same textual instruction but without an image.

| Model | Logical | | Numerical | | Annotations | | | |
| | POL | POC | ALC | LHC | PEP | PRA | EQL | Overall |
|---|---|---|---|---|---|---|---|---|
| Random Baseline | 1.53 | 3.90 | 59.92 | 51.36 | 0.47 | 0.00 | 0.02 | 16.74 |
| *Open Source* | | | | | | | | |
| Molmo (Deitke et al., 2024) | 12.84 | 37.60 | 56.77 | 16.79 | 1.89 | 0.00 | 0.81 | 18.10 |
| Llama32 (Dubey et al., 2024) | 17.36 | 40.67 | 59.46 | 52.08 | 14.59 | 23.58 | 49.91 | 36.81 |
| Qwen2VL (Wang et al., 2024a) | 22.83 | 41.78 | 46.60 | 63.27 | 32.89 | 33.02 | 54.40 | 42.11 |
| Pixtral (AI, 2023) | 26.20 | 60.45 | 47.33 | 51.43 | 29.97 | 38.68 | 58.50 | 44.65 |
| *Closed Source* | | | | | | | | |
| GPT-4omini (Achiam et al., 2023) | 10.52 | 62.95 | 48.84 | 69.51 | 12.22 | 4.72 | 44.77 | 36.22 |
| GPT-4o (Achiam et al., 2023) | 17.10 | 76.88 | 55.63 | 74.39 | 32.18 | 65.09 | 44.75 | 52.29 |
| Claude (Anthropic, 2024) | 26.41 | 74.93 | 42.95 | 70.73 | 34.07 | 71.70 | 66.41 | 55.31 |
| Gemini Flash (Team et al., 2023) | 30.83 | 71.31 | 49.89 | 76.69 | 42.59 | 71.70 | 66.32 | 58.47 |
| Gemini Pro (Team et al., 2023) | 25.14 | 71.31 | 57.96 | 79.05 | 52.37 | 85.85 | 52.32 | 60.57 |

Table 7: Performance (average evaluation score) of different models on Geoperception benchmark tasks. The evaluation score is computed as the ratio of the intersection of predictions ($P$) and ground truth ($G$) to the size of the ground truth ($|G|$): Evaluation score = $\frac{|P \cap G|}{|G|}$, . POL: PointLiesOnLine, POC: PointLiesOnCircle, ALC: AngleClassification, LHC: LineComparison, PEP: Perpendicular, PRA: Parallel, EQL: Equals. As the Random Baseline method, we use GPT-4o-mini, given the same textual instruction but without an image.

| Model | Logical | | Numerical | | Annotations | | | |
| | POL | POC | ALC | LHC | PEP | PRA | EQL | Overall |
|---|---|---|---|---|---|---|---|---|
| Random Baseline | 21.11 | 13.97 | 59.92 | 51.36 | 3.92 | 8.65 | 0.01 | 22.70 |
| *Open Source* | | | | | | | | |
| Molmo (Deitke et al., 2024) | 50.25 | 72.21 | 56.77 | 76.61 | 15.01 | 14.43 | 51.27 | 48.08 |
| Llama32 (Dubey et al., 2024) | 41.43 | 84.60 | 59.46 | 52.22 | 7.43 | 22.21 | 50.56 | 45.42 |
| Qwen2VL (Wang et al., 2024a) | 22.16 | 90.46 | 46.60 | 63.27 | 18.69 | 18.16 | 54.38 | 44.82 |
| Pixtral (AI, 2023) | 36.92 | 80.57 | 47.33 | 51.43 | 11.79 | 21.34 | 57.69 | 43.87 |
| *Closed Source* | | | | | | | | |
| GPT-4o-mini (Achiam et al., 2023) | 57.32 | 90.53 | 48.84 | 69.51 | 18.09 | 24.92 | 44.42 | 50.52 |
| GPT-4o (Achiam et al., 2023) | 49.47 | 89.36 | 55.63 | 74.39 | 20.17 | 31.88 | 44.21 | 52.16 |
| Claude 3.5 Sonnet (Anthropic, 2024) | 48.95 | 88.67 | 42.95 | 70.73 | 11.25 | 32.35 | 65.80 | 51.53 |
| Gemini-1.5-Flash (Team et al., 2023) | 44.36 | 85.33 | 49.89 | 76.69 | 19.22 | 32.19 | 65.85 | 53.36 |
| Gemini-1.5-Pro (Team et al., 2023) | 54.44 | 90.83 | 57.96 | 79.05 | 21.52 | 38.80 | 50.32 | 56.13 |

