# OpenReview forum: "Euclid: Supercharging Multimodal LLMs with Synthetic High-Fidelity Visual Descriptions"
_ICLR.cc/2025/Conference — Submitted to ICLR 2025_

### Official Review · Reviewer_hLZB · 2024-10-16

**Soundness:** 3
**Presentation:** 3
**Contribution:** 3
**Rating:** 3
**Confidence:** 3

**Summary:**

The authors present Geoperception, a VQA dataset sourced from Geometry-3K. Geoperception requires reasoning over "surface-level geometric information" to identify points and line segments that satisfy conditions, and to classify angles. The authors evaluate a variety of existing MLLMs on the dataset and find that they do not perform consistently. Motivated by the results of the evaluation, the authors train their own MLLM, Euclid, on synthetic samples created using modified AlphaGeometry tooling. They also measure the effects of several design choices on their synthetic dataset.

**Strengths:**

1. The writing of the paper is above-average.
2. Diagnostic benchmarks are valuable and relevant to the ICLR community.
2. The results of the visual-tokenizer evaluation are interesting.

**Weaknesses:**

1. Performance on the benchmark seems likely moreso dominated by syntax than reasoning ability: "Certain models, such as GPT-4o-mini (Achiam et al., 2023) and Molmo-7B-D (Deitke et al., 2024), frequently either enumerate all potential components (e.g., all points in a diagram instead of the one on the lines) or every potential answer, leading to their poor accuracy scores." Given the authors have identified this as an issue, it seems difficult to conclude from the investigation that "Current MLLMs struggle to perceive low-level geometry annotations and relationships."

    Some questions are ungrammatical, such as asking "What is the point lying on circle with center G?" when the correct answer includes three points. Some are ambiguous, such as "Which line is longer, AE or ED?" where the answer can not be discerned from the symbols and requires measurement. Others are inconsistently labeled, like alphabetizing segment ends in answers ("CD") but not questions ("What is the line parallel to line EB?") or including duplicate answers ("ZV, VZ").
2. If the task itself had real-world application, the effect of syntax would be of lesser concern. However, "as a focused test bed" "designed to evaluate MLLMs’ ability to accurately perceive surface-level geometric information without requiring complex inference or reasoning" it is far removed from the "real-world applications" used to motivate the work: "spatial understanding for robotics, medical image analysis for accurate diagnosis, quality control in manufacturing to detect subtle defects, autonomous driving systems that rely on exact object localization or distance estimation, and augmented reality applications that demand precise overlay of virtual objects onto the real world."
3. The "detailed empirical exploration of MLLM architecture and training strategies" is disappointingly shallow. The only ablated architecture choice is the visual encoder. In their "investigation," the authors do not ask why the "CNN architecture performs better than ViT," they only present it as a "lesson." The second lesson, that "Tuning the visual encoder is beneficial," should be considered a given when one is evaluating on the training distribution. What is potentially compromised is generalization ability, which is why the ablations should have instead been measured on the evaluation set.
4. The authors' statement that they "demonstrate the limitations of leading MLLMs, and then conduct a comprehensive empirical study to explore strategies for improving *their* performance on geometric tasks" (emphasis added) is misleading. The authors do not improve the performance of existing MLLMs, opting instead to train their own from an existing visual encoder and language model. As such, it is difficult to say that "lessons we learn from this domain can be effectively generalized to a broader set of downstream domains that benefit from high-quality low-level visual perception." Had the authors tried finetuning an existing model, it's plausible that it would not be necessary to introduce a "data curriculum" to enable "models to learn challenging geometry understanding tasks which they fail to learn from scratch."
5. The curriculum training appears irrelevant as the "dataset generation engine can produce infinite samples for exhaustive task-specific training" and "Adaptive Curriculum" performance appears to match "Mixed Shape" at saturation when trained with 150% the samples (Figure 6). Why is it of practical value to "further enhance model efficiency on challenging shapes" and why is it true that "employing a curriculum-based training approach yields much more model potential than direct task training"?
6. The authors' claims are overstated. It is not true that "Euclid significantly outperforms current leading MLLMs," "surpassing the leading MLLMs by a substantial margin." It is outclassed by Gemini-1.5-Pro on one of the four tasks and exceeds it by less than a percent on another. The authors must correctly qualify their claims.

**Questions:**

1. Why are only four out of seven benchmark tasks used for evaluation of the proposed model (Table 4)?
2. In the related work, the authors write that "VDLM (Wang et al., 2024b) transcribes raster images into vector graphics and uses LLMs to reason over the SVG code. They find that although SVG code is not straightforward to understand, using LLMs to reason over SVG is consistently more effective than directly using MLLMs on original raster images." If reconstructing prior to reasoning is "consistently more effective," why is it not evaluated against? Why is it important that Geoperception be solved through intermediate-free VQA?
3. The authors state their task is "straightforward for humans" but do not provide quantitative comparison. Have they measured this?

Further comments:
1. The title does not seem very representative of the paper content and appears overly sensational. The authors are suggested to change it to remove mention of Euclid, which the authors have not demonstrated to be generally applicable outside the benchmark, and focus on Geoperception which seems to be the real contribution of the work.
2. The authors write "For our Euclid model, we include all geometry shape code used for training, along with demonstration diagrams and pseudocode for generating training questions and answers," but it does not appear that supplementary materials were submitted.

---

> ### Author Response · Authors · 2024-11-25
> **Official Comment by Authors (1/3)**
>
> Thank you for your suggestions and insightful comments. We really appreciate your effort in reviewing the paper. We address your questions and comments below:
>
>
> ### **Concern about whether the performance on Geoperception is dominated by syntax error.**
>
> To improve clarity, in Appendix B we show the full prompts given when evaluation models on Geoperception, which include both instructions in addition to the original question.  For example, in the instructions, we separately give the answer template for both single answer point and multiple answer points. We observed that all evaluated models adhered to these instructions, indicating no difficulty in understanding verbal instructions and questions. For length comparison tasks, during construction, we make sure no annotations are made on the lines in the question. Regarding label consistency, we randomly shuffle the order of the letters for segments instead of alphabetizing them. During evaluation, ‘ZV’ and ‘VZ’ are considered as one ground truth answer, we include both of them for matching purposes. Given the detailed and clear instructions that we give to the model, and the fact that most of the MLLMs achieve reasonable performance, suggest that ‘frequently enumerating all potential components’ should be considered as a visual perception weakness rather than syntax error. We will clarify the above in our updated paper.
>
>
> ### **Concern about real-world application.**
>
> We acknowledge that our designed dataset differs from real-world downstream applications; geometry perception, however, is a fundamental prerequisite for low-level visual perception in practical scenarios—a model without reliable performance on these simple and straightforward tasks could not be trusted to perform low-level visual perception in more complex settings. Moreover, geometry understanding is a focused and controllable domain where datasets can be programmatically generated and different aspects, such as task complexity, can be systematically manipulated. This enables us to better diagnose the root causes of the model's shortcomings and explore strategies for addressing them effectively.
>
>
> ### **Concern about empirical exploration.**
>
> In this study, we focus on low-level geometric visual perception, deliberately avoiding tasks that require additional reasoning. In this context, we consider the transcription of visual information into the language space to be the most critical component. Consequently, we emphasize the selection of visual encoders. Regarding the superior performance of CNN architectures, we hypothesize that CNNs extract visual features globally, effectively preserving low-level visual features. In contrast, ViT architectures split images into discrete patches, making it more challenging to retain the original low-level visual information. We will incorporate this explanation into the next version of the paper.
>
> Following your suggestion, we conducted an experiment to study the effect of LLM size, where we train 0.5B, 1.5B and 3B models with the same other model components and the same dataset. We evaluated both the training loss and testing accuracy curves. The results indicate that scaling up the LLM size does not lead to significant improvements. This observation further supports our choice of using a 1.5B LLM. We have included these findings, along with the training loss and testing accuracy curves, in Figure 18 of the updated paper version and plan to move them to the main text in the next version.
>
> Notably, the trends in the training and testing accuracy curves are well-aligned. We hypothesize that this alignment arises because the training and test datasets are generated by the same dataset engine and share the same distribution. Therefore, the training loss trends are representative for comparing different training strategies. That said, we fully agree that evaluating on a separate test set provides additional value. In response, we are re-running the entire empirical study, performing test set evaluations at every 50 training steps, and will include these results in the updated version of our paper.

---

> ### Author Response · Authors · 2024-11-25
> **Official Comment by Authors (2/3)**
>
> ### **The value of curriculum learning.**
>
> As shown in Figure 6, directly training the model on Shape 3—the most complex shape—leads to non-convergence. In contrast, curriculum learning, including mixed-shape training (illustrated as spontaneous curriculum learning in Figure 5), achieves convergence with the same amount of training data. This transition from non-convergence to convergence demonstrates a significant improvement in the training process, enabling better utilization of the training data and unlocking greater potential for model performance.
> Furthermore, we find that adaptive curriculum is more efficient than mixed-shape training to be practically valuable. This is because: 1. For many tasks, such as real-world low-level visual perception, training datasets are often constrained, making it critical to explore and identify optimal training strategies. 2. Although our dataset generation engine can theoretically produce unlimited task-specific training samples, compute resources are typically a limiting factor—particularly when training large models or scaling the method to broader and more diverse data distributions.
>
> > It is not true that "Euclid significantly outperforms current leading MLLMs," "surpassing the leading MLLMs by a substantial margin." It is outclassed by Gemini-1.5-Pro on one of the four tasks and exceeds it by less than a percent on another.
>
> Thanks for pointing this out. It is true that our model does not outperform the leading MLLMs across all tasks, instead our best all-in-one Euclid model surpasses Gemini-1.5-pro by 10.21% by average of four tasks. We have clarified the claims in the updated version of the paper. (Line 471-480)
>
> > Why are only four out of seven benchmark tasks used for evaluation of the proposed model (Table 4)?
>
> In our Euclid testing, the objective is to evaluate whether the low-level visual perception capabilities learned from synthetic data training can effectively generalize to shapes in real-world geometry problems (i.e., Geoperception). We focus on the four primitive tasks because:
> 1. As shown in Table 3, current MLLMs struggle significantly with these tasks compared to the three annotation-based tasks.
> 2. These tasks do not require human-defined annotations on geometric diagrams, providing a clean and unambiguous testbed to assess the model's ability to generalize geometric shapes. In contrast, the three annotation-based tasks rely heavily on manually designed diagram annotations (e.g., arrows pointing to angles or specific styles to denote parallel lines), introducing undesirable gaps between our training datasets and the Geoperception benchmark.
>
> Nonetheless, to address your concerns, we extended our geometry generation engine to produce diagrams for the annotation-based tasks, including segment length, angle measure, parallelism, and perpendicularity, with each task having a single consistent annotation style. Instead of training Euclid on separate tasks, both models are jointly trained on all seven tasks. We present the updated results below and will include them in the revised version of the paper:
> | Model                       | POL   | POC   | ALC   | LHC   | PEP   | PRA   | EQL   | Overall |
> |-----------------------------|-------|-------|-------|-------|-------|-------|-------|---------|
> | Random Baseline            | 0.43  | 2.63  | 59.92 | 51.36 | 0.25  | 0.00  | 0.02  | 16.37   |
> | Pixtral-12B [Pixtral]      | 22.85 | 53.21 | 47.33 | 51.43 | 22.53 | 37.11 | **58.45** | 41.84   |
> | Gemini-1.5-Pro [Gemini]    | 24.42 | **69.80** | 57.96 | 79.05 | **39.60** | **77.59** | 52.27 | 57.24   |
> | **Euclid-ConvNeXt-Large**  | 72.14 | 50.02 | 84.63 | 84.43 | 37.21 | 58.41 | 21.44 | 58.33   |
> | **Euclid-ConvNeXt-XXLarge**| **80.20** | 56.36 | **87.10** | **85.58** | 39.06 | 58.81 | 27.23 | **62.05** |
>
> In general, introducing geometric annotations into the training dataset improves Euclid's performance on most of the promotive tasks (except for PointLiesOnCircle). We hypothesize that this improvement is due to increased model robustness resulting from training on a more diverse dataset. However, for the three annotation-based tasks, Euclid’s performance does not surpass the current leading MLLMs. We attribute this to the annotation style gap between our synthetic training data and the Geoperception benchmark, as previously explained.

---

> ### Author Response · Authors · 2024-11-25
> **Official Comment by Authors (3/3)**
>
> ### **Why is it important to solve Geoperception intermediate-free VQA, instead of methods like VDLM?**
>
> Thank you for bringing this up. We agree that alternative methods, such as VDLM, could also be applied to address geometry perception problems. However, the focus of this paper is not on identifying a specific solution for Geoperception tasks. Instead, we use these tasks as a representative testbed to investigate the design principles of MLLMs and their ability to effectively learn low-level visual perception concepts. This exploration provides insights that extend beyond geometry perception to general-purpose and domain-specific MLLM training.
> Furthermore, as noted in the VDLM paper, LLMs cannot directly interpret SVG code. VDLM addresses this by training a model to convert SVG code into a predefined language. However, this predefined language is highly constrained and can represent only very simple shapes (see VDLM paper, Page 24 https://arxiv.org/pdf/2404.06479#page=24 for reference—characters, for example, are not supported).
>
> In contrast, our empirical study demonstrates that synthetic data training enables MLLMs to generalize low-level perception abilities to a wider range of geometric shapes. This approach directly contributes useful insights into designing MLLMs for low-level visual perception, both for general-purpose applications and specific domains. The Euclid testing results further validate this by showing that our model, trained on synthetic data, successfully generalizes its learned low-level visual abilities to diverse geometry shapes.
>
> > The authors state their task is "straightforward for humans" but do not provide quantitative comparison. Have they measured this?
>
> Thanks for bringing this up. To address your concern, we have randomly chosen 20 examples from each task from Geoperception, and performed a human evaluation. The result showed that all 140 datasets are quickly and correctly solved by humans, demonstrating that the Geoperception benchmark is very straightforward for humans. We will add this human experiment in the updated version of the paper
>
> > The authors write "For our Euclid model, we include all geometry shape code used for training, along with demonstration diagrams and pseudocode for generating training questions and answers," but it does not appear that supplementary materials were submitted.
>
> The geometry shapes and demonstrations are included in the appendix C.2, and the pseudocode for generating the training questions and answers are included in appendix C.1.

---

> > ### Comment · Reviewer_hLZB · 2024-11-26
> >
> > ### W1
> > The authors previously noted "all [baseline] models fall short of achieving 30% accuracy on the PointLiesOnLine task and do not outperform the text-only GPT-4o mini model in AngleClassification task." This is not aligned with their rebuttal justification that "most of the MLLMs achieve reasonable performance." The authors' conclusion that the models' ability to answer one question type with correct formatting implies they have "no difficulty in understanding verbal instructions and questions" is not well-founded. It has not been demonstrated that "top models such as Gemini-1.5-Pro struggle significantly" as a result of "visual perception weakness rather than syntax error." As suggested by reviewer HWJq, the authors should have formatted the questions as multiple choice to avoid what they describe in the following: "Certain models, such as GPT-4o-mini (Achiam et al., 2023) and Molmo-7B-D (Deitke et al., 2024), frequently either enumerate all potential components (e.g., all points in a diagram instead of the one on the lines) or every potential answer, leading to their poor accuracy scores."
> > ### W5
> > The authors' characterization of "mixed-shape training" as "spontaneous curriculum learning" is non-standard and is not aligned with literature. As it stands, it has not been demonstrated that "a data curriculum enables models to learn challenging geometry understanding tasks which they fail to learn from scratch" and the claim must be removed from the paper.
> > ### Q1
> > The authors' decision to evaluate their model on only a subset of the proposed benchmark raises concern. When the full benchmark is included, the overall quantitative edge of the proposed model over the top-performing zero-shot baseline decreases from 14.29% to 4.81%. If "top models ... struggle significantly," it cannot be said that "Euclid achieves strong performance."
> >
> > It is unclear why the baselines succeed less than half as frequently on the PointLiesOnLine task than on the similar PointLiesOnCircle evaluation, while a reverse relationship is observed with Euclid. If the PointLiesOnLine task is then removed, the baseline difference flips to -3.69%. The underwhelming quantitative improvement, coupled with the authors' identified syntax issues, casts doubt on the significance of the results.

---

> > > ### Comment · Reviewer_hLZB · 2024-11-26
> > >
> > > The paper is not what the authors present it as, and extensive restructuring -- rather than rebuttal-period band-aids -- is necessary to support the claims made. The title is not descriptive of the work (C1). The authors do not "explore strategies for improving [leading MLLMs'] performance on geometric tasks," instead opting to train their own model from separate existing visual encoders and language models. It has not been demonstrated that "lessons we learn from this domain can be effectively generalized to a broader set of downstream domains that benefit from high-quality low-level visual perception." This is due first to the authors' decision to evaluate ablations solely on the synthetic training domain ("the training and test datasets are generated by the same dataset engine and share the same distribution") (W3) and second because the authors do not attempt to finetune an existing model, for which the introduced curriculum may have had no effect (W4). Quantitative results are further not aligned with the authors' claims ("top models ... struggle significantly"; "Euclid achieves strong performance") (W6), with zero-shot baselines outperforming the authors' model (Euclid) in the majority of evaluation tasks on the proposed benchmark (Geoperception) and having only a quantitative edge of 4.81% in the aggregate (Q1) which seems more likely due to acknowledged prompting issues than failures of visual perception (W1). A visual diagnostic should not be hard because it is hard to prompt.
> > >
> > > The authors' rebuttal failed to address concerns originally outlined and has raised others. I am decreasing my rating from 5 to 3 accordingly.

---

> > > > ### Author Response · Authors · 2024-11-27
> > > >
> > > > ### **W1**
> > > >
> > > > As evidenced by our manual experiment in Line 197-199, GPT-4o-mini knows well about the existence of all points presented on the diagram. In Geopercetion, most of our questions are to pick a subset from a full set (e.g. pick the points on a certain line from all points), given that the GPT-4o-mini knows well about all existing points, the task is essentially multiple choice question with multiple correct answers. We understand that the reviewer thinks the behavior “frequently either enumerates all potential components or every potential answer” is due to improper prompting. To further address your concern, we run GPT-4o-mini again on the PointLiesOnLine task, but giving it **full text-based logical annotations from Geometry-3K** and exactly the same question and instructions. If syntax error dominates the model’s behavior, we should see a similar performance on this setting, since GPT-4o-mini will still enumerate all potential answers. However, we find that GPT-4o-mini's performance on PointLiesOnLine reaches **24.40%**, it suggests that the model can make better predictions with the same instructions. We provide an example of the raw logical annotation from Geometry-3K below:
> > > >
> > > > {
> > > >   "text_logic_form": ["Find(y)"],
> > > >   "dissolved_text_logic_form": ["Find(y)"],
> > > >   "diagram_logic_form": ["Equals(LengthOf(Line(C, D)), 10)", "Equals(LengthOf(Line(B, C)), x)", "Equals(LengthOf(Line(B, D)), 4)", "Equals(LengthOf(Line(A, B)), y)", "Equals(LengthOf(Line(A, D)), z)", "PointLiesOnLine(B, Line(A, D))", "Perpendicular(Line(A, C), Line(D, C))", "Perpendicular(Line(B, C), Line(A, B))"],
> > > >   "line_instances": ["AB", "AC", "AD", "BC", "BD", "CD"],
> > > >   "point_positions": {
> > > >     "A": [1.0, 112.0],
> > > >     "B": [253.0, 113.0],
> > > >     "C": [251.0, 2.0],
> > > >     "D": [301.0, 112.0]
> > > >   },
> > > >   "circle_instances": ["”]
> > > > }
> > > >
> > > >
> > > >
> > > > ### **W5**
> > > >
> > > > In our work, we did demonstrate that a data curriculum enables models to learn challenging geometry understanding tasks that they fail to learn from scratch. This conclusion is supported by the analysis presented in Lines 397–406. Specifically, Figure 3 highlights that ***all models fail to converge when training solely on Shape 3, whereas mixed training achieves convergence despite utilizing the same amount of Shape 3 data***. To further investigate, we separately plotted the training loss for Shapes 1, 2, and 3 during mixed-shape training, as shown in Figure 5. The results reveal ***an apparent plateau*** in the loss curve for Shape 3 before approximately 20K training samples. This suggests that the model initially struggles with Shape 3 but gradually improves after leveraging prior learning from simpler shapes. These findings strongly indicate that learning low-level geometric concepts from easier shapes facilitates the model’s ability to understand more complex shapes. (spontaneous curriculum learning) We consider this result a key contribution of our paper, as it provides clear evidence that in learning low-level visual perception, the simplicity of initial training tasks is critical for enabling the model to progress toward more complex tasks. In contrast, many general multimodal training frameworks lack justification for whether the initial low-level perception tasks are sufficiently simple for effective learning.
> > > >
> > > >
> > > >
> > > > ### **Q1**
> > > > Please note that when we introduce the annotation tasks Euclid’s performance on four primitive tasks increased by at most 10.62% percent, surpassing the best proprietary MLLM, Gemini-1.5-pro by 19.50%. As we previously mentioned, the three tasks that we did not include originally rely heavily on arbitrary human-defined annotations. We attribute Euclid’s performance to the annotation style gap between our synthetic training data and the Geoperception benchmark, as previously explained.
> > > > Regarding the overall performance, please note that the Euclid model is 1. Only fine tuned on synthetic data containing simple geometry shapes and annotation types. 2. Use only 1.5B LLM, which is hundreds of times smaller than some proprietary models. We believe this is a ***noteworthy achievement*** given that Euclid’s performance surpasses the best proprietary models by 4.81% overall and by 19.50% in primitive tasks.
> > > >
> > > > After including the three additional annotation tasks, we will make sure to change the term in the next version of the paper accordingly to *Euclid achieves strong performance on primitive tasks while not outperforming the current leading MLLMs in annotation recognition tasks.*

---

> > > > > ### Author Response · Authors · 2024-11-27
> > > > >
> > > > > ### **Concern about not training an existing ready-to-use MLLM.**
> > > > >
> > > > > In our experiments, we adopt the high-level design framework of LLaVA-1.5, which includes a pre-trained vision encoder, an MLP multimodal connector, and a pre-trained large language model. The key distinction is that we train our MLLMs using datasets generated by our geometry dataset engine, whereas other MLLMs are typically trained on a combination of general-purpose multimodal instruction-tuning datasets.
> > > > >
> > > > > Existing MLLMs differ significantly in several aspects, such as architectural choices (e.g., vision encoder, LLM, and multimodal connector), training datasets (many of which are inaccessible or unspecified), and training hyperparameters. Consequently, training existing MLLMs on our task does not allow for a rigorous evaluation of training strategies or architectural design. Moreover, certain architectural configurations we investigate—such as using ConvNeXt as the vision encoder—are uncommon among current MLLMs. By training our own model, we gain greater flexibility and ensure a fair comparison.
> > > > >
> > > > > Additionally, as demonstrated in Table 3, existing MLLMs, especially open-sourced ones, perform poorly on Geoperception tasks. This finding indicates that, even after extensive general multimodal instruction tuning, these models still exhibit limited capabilities in fundamental low-level visual perception. Thus, we argue that tuning existing MLLMs would yield results which are similar to those obtained with our experimental setup, without addressing the underlying limitations.

---

### Official Review · Reviewer_HWJq · 2024-11-03

**Soundness:** 2
**Presentation:** 1
**Contribution:** 2
**Rating:** 5
**Confidence:** 3

**Summary:**

The paper proposes a new benchmark, Geoperception, sourced from Geometry-3K, to evaluate MLLMs' ability to perform geometric perception. The dataset is in QA form, and the benchmark has seven tasks. Most are about element relationships in geometric images, such as points, lines, and angles. The author also proposes a method called Euclid, a fine-tuned MLLM that may outperform other MLLMs.

**Strengths:**

Geometric perception benchmark may seem a novel topic to me. The benchmark can contribute to the MLLM community.

**Weaknesses:**

- Overall, this paper devotes a significant amount of space to introducing background information and other methods, while providing relatively fewer details about its own proposed approach. I think the amount of work is below the average level for ICLR. Therefore, I believe the contributions of this paper are insufficient for acceptance at a top-tier conference like ICLR.

- The proposed task is sourced from Geometry-3K and seems to be easier than it. Since Geometry-3K contains some calculations, while this benchmark is mainly about some relationships between elements, such as whether point A is on Line A. I know the paper is trying to explore the geometric perception topic, but Geometry-3K also needs some level of perception, or models cannot do the harder calculation.

- For baselines, why not directly fine-tune some MLLMs on Geoperception, to compare with the method of Euclid?

- If an MLLM has been fine-tuned on this perception dataset, will it gain the ability of other types of perception, such as real images or medical images?

**Questions:**

- Why not consider the multiple-choices form for question answering?

- Is it possible to transform geometric figures into SVG code, just as VDLM, to evaluate the performance of the OpenAI-o1 model? Moreover, the authors may add some evaluation on some geometry MLLMs or math MLLMs.

---

> ### Author Response · Authors · 2024-11-25
> **Official Comment by Authors (1/2)**
>
> Thank you for your suggestions and insightful comments. We really appreciate your effort in reviewing the paper. We address your questions and comments below:
>
> > Overall, this paper devotes a significant amount of space to introducing background information and other methods, while providing relatively fewer details about its own proposed approach.
>
> Thank you for your feedback. To better clarify, we reiterate our contribution of this paper below (as we feel a lot of what you might view as background information, we view as our contributions):
> 1. **Geoperception Benchmark**: We introduce a novel benchmark specifically designed to evaluate MLLMs' low-level geometric perception, revealing critical shortcomings in current models. This benchmark offers a clear testbed for future work on precise visual perception.
> 2. **Empirical Study on Synthetic Data Engine**: We conduct a detailed design study of architectural and training strategies, supported by a high-fidelity synthetic data engine. This enables us to provide actionable insights, such as the effectiveness of curriculum-based training, with implications for general-purpose low-level visual perception.
> 3. **Euclid Model**: Leveraging these insights, we propose the Euclid model series, which significantly outperforms current leading MLLMs on many tasks and by average (e.g., up to 54.52% improvement over Gemini-1.5-Pro on certain perception tasks) despite being trained only on synthetic data and having only 1.5B-parameter LLM.
>
> ### **Relationship with Geometry-3K**
>
> We fully agree that the Geometry-3K dataset is inherently more complex than Geoperception. However, as the reviewer agrees, the foundational low-level visual perception of MLLMs should first be sufficiently robust, as a precursor to performing more-complex calculations or reasoning. The primary objective of proposing Geoperception is to isolate and rigorously evaluate this fundamental aspect of perception.
>
> Our evaluation on Geoperception, as shown in Table 2, demonstrates that all leading MLLMs encounter significant difficulties in basic geometric perception tasks. The scope of this paper is specifically to tackle this critical challenge.
>
> > For baselines, why not directly fine-tune some MLLMs on Geoperception, to compare with the method of Euclid?
>
> Thanks for bringing this up. As a benchmark dataset, Geoperception’s volume (Table 1)  is not enough for a MLLM training, which typically needs at least tens of thousands of training instances. Furthermore, we were aiming to keep Geoperception data as held-out test data (so as not to "cheat" nor contaminate our training set with test data).
>
> To further address your concern, we tested Cambrian-1[2] on our Geoperception dataset, which is reported to be trained on Geo-170K, a geometry multimodal instruction tuning dataset built on the logical annotation of Geometry-3K, having the same source with our Geoperception. The testing result below shows that Cambrian-1 also faces challenges in our Geoperception task, despite being trained on the dataset having the same images and augmented text annotations:
>
>
> | Model                          | PointLiesOnLine | PointLiesOnCircle | AngleClassification | LineComparison | Perpendicular | Parallel | Equals |
> |--------------------------------|-----------------|-------------------|---------------------|----------------|---------------|----------|--------|
> | Random Baseline  | 0.43            | 2.63              | 59.92               | 51.36          | 0.25          | 0.00     | 0.02   | 16.37 |
> | Cambrian-1-8B  | 15.14           | 28.68             | 58.05               | 61.48          | 18.78         | 26.18    | 31.04  | 34.19 |
> | Pixtral-12B        | 22.85           | 53.21             | 47.33               | 51.43          | 22.53         | 37.11    | 58.45  | 41.84 |
> | Gemini-1.5-Pro | 24.42           | 69.80             | 57.96               | 79.05          | 39.60         | 77.59    | 52.27  | 57.24 |

---

> ### Author Response · Authors · 2024-11-25
> **Official Comment by Authors (2/2)**
>
> ### **Generalizing to Real World Images**
>
> We totally agree that gaining the low-level visual perception ability of other types of images is very important, and is a future goal in this line of work. However, we think the study of generalization ability merits its own paper so we leave it to future works. In this paper, our scope is to systematically find out the problem in today’s MLLMs, and explore optimal training architecture and data composition choices to learn such tasks effectively and efficiently.
>
> > Why not consider the multiple-choices form for question answering?
>
> Thanks for the suggestion. In most of the questions in Geoperception, we expect a list of ground truth answers. For example, all points lying on the same line. To this end, we choose to evaluate the model’s performance smoothly (line 224), rather than using a binary evaluation metric. While we fully agree that using the format of multiple choice questions can reduce the ambiguity and make the evaluation more rigid. We observed that all evaluated models adhered to the detailed instructions (outlined in Appendix B) indicating no difficulty in understanding verbal instructions and questions.
>
> > Is it possible to transform geometric figures into SVG code, just as VDLM, to evaluate the performance of the OpenAI-o1 model? Moreover, the authors may add some evaluation on some geometry MLLMs or math MLLMs.
>
> Thanks for the suggestion. We agree that methods like VDLM is another strategy for solving similar geometry problems. However, instead of finding alternative ways to solve these geometry problems, the goal in this paper is to use the geometry perception as a representative testbed to explore the design of MLLMs so that they can effectively learn the low-level visual perception concepts. Furthermore, in the VLDM paper, they noted that LLMs are not able to directly understand the SVG codes, so they trained a model to transform the original SVG code into a predefined language. Under this language, however, only very simple shapes can be represented, (se​​e https://arxiv.org/pdf/2404.06479#page=24 for reference, even characters are not covered)..
>
> Following your suggestion, we run geometry MLLMs on Geoperception task to assess their performance. However, we find that they fail to follow instructions properly. Here is one example of G-LLaVA[1]: (the input image is similar to the images we present in Figure 2)
>
> Input: <image> What is the point lying on line RB?
>
> Output: Solution: Since angle BAC = 30° and angle B = 45°, we have angle B = 15°. Since AC = BC, we have angle C = angle BAC = 30°. Therefore, angle BAC = angle C = 30°. Thus, triangle ABC is an equilateral triangle, so AB = BC. Since AB = BC, we have AB = AC. Therefore, the answer is B.
> Answer:B
>
> Note that in this example, instead of following instructions properly and identifying the point lies on a certain line, the model outputs an irrelevant geometry problem solution.
> We hypothesize such behavior is sourced from the limited training dataset diversity of the geometry-oriented models, which are typically only trained on geometry problem solving.
>
>
> [1] Gao, J., Pi, R., Zhang, J., Ye, J., Zhong, W., Wang, Y., ... & Kong, L. (2023). G-llava: Solving geometric problem with multi-modal large language model. arXiv preprint arXiv:2312.11370.
>
> [2] Tong, S., Brown, E., Wu, P., Woo, S., Middepogu, M., Akula, S. C., ... & Xie, S. (2024). Cambrian-1: A fully open, vision-centric exploration of multimodal llms. arXiv preprint arXiv:2406.16860.

---

> > ### Comment · Reviewer_HWJq · 2024-11-25
> > **post-rebuttal**
> >
> > I would like to thank the authors for their detailed comments. Some of my concerns have been solved, but I still feel confused about the proposed benchmark. Although the authors have proposed an interesting idea, they may overstate some of the contributions in my opinion. I believe the proposed benchmark and the empirical study will exactly contribute to the community, while the existing issue remains. We still have difficulty overcoming the constraints of real tasks such as medical image analysis and robotics proposed in the Abstract. Moreover, as declared by the authors, the data volume is hard for MLLMs to fine-tune, so the contribution is further limited. Thus I would like to set my score to five.

---

> > > ### Author Response · Authors · 2024-11-26
> > >
> > > Thanks for the response, we are pleased that some of your concerns have been solved and we feel encouraged that you raise your score and think our study will contribute to the community!
> > > In the following, we would like to response to some of the remaining concerns:
> > >
> > > > We still have difficulty overcoming the constraints of real tasks such as medical image analysis and robotics proposed in the Abstract.
> > >
> > > We fully acknowledge that the challenge of low-level visual perception remains significant across various domains, including medical image analysis and robotics, which is evidenced by many recent studies, such as BLINK [1]. Based on our findings, we attribute this issue primarily to two factors:
> > > - Lack of high-fidelity multimodal training dataset.
> > > - Limitations in existing training strategies.
> > >
> > > As an initial attempt addressing this challenge, we choose geometry understanding as the focus of our study for the following reasons:
> > >
> > > - It serves as the foundational element of low-level visual perception in many other domains.
> > > - Unlike real-world images, geometric shapes can be programmatically defined and generated, enabling better scalability and controllability.
> > >
> > > Although the existing issue remains, the lesson we learned from this paper can be easily applied to other domains. For example, one might find with same training data, using CNN architectures or introduce data curriculum in robotic MLLMs could improve their ability in executing the tasks requiring precise visual perception.
> > >
> > > > Moreover, as declared by the authors, the data volume is hard for MLLMs to fine-tune, so the contribution is further limited.
> > >
> > > The Geoperception benchmark is designed to measure MLLMs' low-level perception capabilities in real world 2D geometric contexts rather than to serve as a training dataset. As demonstrated in Section 4, our dataset generation engine (generating synthetic data which is different from Geoperception) is capable of producing sufficient data for fine-tuning MLLMs. Moreover, as evidenced in Section 5, the fine-tuned models exhibit strong generalization to real-world geometric diagrams (Geoperception), demonstrating the practical utility of our approach.
> > >
> > > Thanks again for your recognition and effort, they are very helpful in improving this paper! We hope these could address some of your remaining concerns. Please let me know if you have further questions!
> > >
> > > [1] Fu, X., Hu, Y., Li, B., Feng, Y., Wang, H., Lin, X., ... & Krishna, R. (2025). Blink: Multimodal large language models can see but not perceive. In European Conference on Computer Vision (pp. 148-166). Springer, Cham.

---

> > > > ### Author Response · Authors · 2024-11-30
> > > >
> > > > Dear reviewer HWJq:
> > > >
> > > > Thank you again for your insightful review! We hope our response has addressed your concerns. And we are happy to engage further if you have any additional questions or suggestions.

---

### Official Review · Reviewer_E5Tw · 2024-11-04

**Soundness:** 3
**Presentation:** 3
**Contribution:** 2
**Rating:** 6
**Confidence:** 4

**Summary:**

This paper addresses the limitations of MLLMs in low-level visual perception. Specifically,  It introduces a benchmark, Geoperception, to evaluate MLLMs' ability to transcribe 2D geometric information.  Effective strategies for enhancing model performance are identified, including high-fidelity synthetic data and a data curriculum approach. It also develops a new family of models, named Euclid, optimized for geometric perception, which significantly outperforms existing models, demonstrating strong generalization to novel geometric shapes.

**Strengths:**

1. The paper is well-written and easy to follow.
2. The paper conducts extensive experiments to systematically explore the impact of geometric shapes of increasing complexity and identifies some key lessons.
3. The paper shows that the Euclid model trained on high-fidelity synthetic data exhibits strong generalization capabilities, especially in real-world geometric shape understanding tasks, significantly surpassing existing leading MLLMs.

**Weaknesses:**

1. Concerns about Data Filtering with MLLM. Since Geoperception is a new benchmark intended to evaluate the precise geometric perception capabilities of MLLMs, and given that these models exhibit limited geometric understanding, it appears unreasonable to utilize gpt-4o-mini for data filtering purposes.
2. Limited Comprehensive Geometric Understanding. The results in Table 4 shows the performance limitations of Euclid on multiple tasks, such as POC and LHC. They may be linked to the benchmark data distribution. It would be beneficial for the authors to conduct further experiments aimed at enhancing this aspect.

**Questions:**

Please refer to the weaknesses above.

---

> ### Author Response · Authors · 2024-11-25
>
> Thank you for your suggestions and insightful comments. We really appreciate your effort and support of the paper’s publication. We address your questions and comments below:
>
> ### **Concerns about Data Filtering with MLLM**
>
> Sorry for the confusion, we only use GPT-4o-mini to verify the existence of points, which is much simpler than answering the training question. To verify the reliability of GPT-4o-mini, we also manually checked 100 examples of its annotation and found only 2 mistakes that the model made (line 197-199).
>
> ### **Limited Comprehensive Geometric Understanding**
>
> Thanks for bringing this up. As is discussed in our error analysis (line 491-495), the images in Geoperception are highly annotated, while our training dataset has only simple geometry shapes and point characters. To make Euclid perform more robustly and address your concern, we have extended our geometry generation engine to enable producing diagrams with annotations, including segment length, angle measure, parallelism, and perpendicularity, with each task having a single consistent annotation style. We then randomly integrate annotations throughout our training dataset. Instead of training Euclid on separate tasks, both models are jointly trained on all seven tasks. We present the updated results below and will include them in the revised version of the paper:
>
> | Model                          | POL   | POC   | ALC   | LHC   | Avg     |
> |--------------------------------|-------|-------|-------|-------|---------|
> | Random Baseline               | 0.43  | 2.63  | 59.92 | 51.36 | 28.59   |
> | Pixtral-12B [Pixtral]         | 22.85 | 53.21 | 47.33 | 51.43 | 43.71   |
> | Gemini-1.5-Pro [Gemini]       | 24.42 | **69.80** | 57.96 | 79.05 | 57.81   |
> | Euclid-ConvNeXt-Large         | 59.52 | 66.18 | 71.41 | 74.96 | 68.02   |
> | Euclid-ConvNeXt-XXLarge       | 55.22 | **70.65** | 66.85 | 74.03 | 66.69   |
> | Euclid-ConvNeXt-Large (improved)         | 72.14 | 50.02 | 84.63 | 84.43 | 72.81   |
> | **Euclid-ConvNeXt-XXLarge (improved)**  | **80.20** | 56.36 | **87.10** | **85.58** | **77.31** |
>
> In general, introducing geometric annotations into the training dataset improves Euclid's performance on most of the promotive tasks (except for PointLiesOnCircle) due to increased model robustness resulting from training on a more diverse dataset.

---

> > ### Author Response · Authors · 2024-11-30
> >
> > Dear reviewer E5Tw:
> >
> > Thank you again for your insightful review! We hope our response has addressed your concerns. And we are happy to engage further if you have any additional questions or suggestions.

---

### Official Review · Reviewer_2bjH · 2024-11-05

**Soundness:** 2
**Presentation:** 3
**Contribution:** 3
**Rating:** 6
**Confidence:** 4

**Summary:**

- The paper studies multimodal large language models (MLLMs) on the task of low-vele visual perception — particularly, testing and improving MLLMs for tasks related to Euclidean geometric concepts like points, lines, and angles.
- The first part of the study highlights significant failures of the recent open-sourced and closed-sourced MLLMs including GPT4o, Gemini, and Molmo.
- In the second part, the authors trained the MLLM on a very simple generated synthetic geometric data, which is composed of only 3 shapes (but multiple questions/answers per shape) for each geometric concept/axiom.
- They also introduce a geometry concept benchmark, named Geoperception, which is a filtered version of Geometry-3K corpus (Lu et al., 2021), as a test-bed to support the above two points.

**Strengths:**

- The paper studies an important and open topic of the application of MLLMs on geometric problem datasets. Furthermore, comparisons of prior SOTA MLLMs and their failures further enhance the importance of the paper.
- The release of the filtered GeoPerception benchmark will support future works.
- Achieves 3x performance boost compared to Gemini1.5-Pro on `PointsonLines` questions and roughty +15% on average.
- Paper writing is overall clear and related works are well covered.

**Weaknesses:**

- Clarification regarding sampling multiple answers per question
    - "Molmo predicts every potential answer, leading to their poor accuracy scores" – If this means, that instead of only answering the required/asked points on the line, Molmo outputs all the points on the line –– this seems less critical. It would be great if the authors could also share the accuracy if the prediction is a superset of the required answer.
    - How are multiple possible answers handled if some answers are incorrect, but there is at least one correct answer? As per evaluation score formulae, score = |Prediction| / |Ground Truth|. It will be again good to know the score of each MLLM if the formulae would have been score=1 if P is a subset of G. This will help understand if the accuracy of MLLMs can be enhanced by non-deterministic nucleus sampling.
    - On the above point, how many solutions are sampled for each question? Is any form of nucleus sampling performed i.e. sampling multiple answers per question? Please provide details for the same.

- What does it mean to compose a dataset of only 3 shapes? Were all the training and the validation question-answer pairs for training generated only from 3 shape instances? Clarification on the differences between the terminology "shapes" and geometric question/answer pairs would be great.

- Following the above question, it might be important to increase the dataset by sampling more shapes from the geometric shape generation engine and then training on that dataset.

- Comparison against a baseline that performs supervised fine-tuning of existing MLLMs rather than training the vision part of MLLMs (fine-tuning vision backbone and adapter) from scratch.

- Furthermore, it seems that the generated training dataset is relatively quite small. Comparison among different convolution and ViT-based image backbones might not be optimal under such small dataset settings. It would be nice to either test this on larger datasets or be subtle of this observation made. The same might be true for the training data curriculum.

- Rather than designing a human-designed curriculum (based on what human finds complex), doing negative-hard mining might be a strategy to design a curriculum automatically. It would be great if the authors could compare against such a strategy.

**Questions:**

- My main concerns are regarding the small size of the dataset (any clarifications on the true train dataset size), or results on enhanced dataset size would be appreciated. Conclusions made on such a small dataset cannot be accepted to scale with dataset size.
- It would be great if the authors could share results from the requested baseline (supervised fine-tuning of existing MLLM) and automatic curriculum training based on negative-hard mining.

If my main concerns on the small training dataset size are addressed, I would be happy to increase my rating.

---

> ### Author Response · Authors · 2024-11-25
> **Official Comment by Authors (1/2)**
>
> Thank you for your suggestions and insightful comments. We really appreciate your effort and support of the paper’s publication. We address your questions and comments below:
>
> ### **Concerns About Training Dataset Volume**
>
> In this paper, we adopt the standard configuration for visual instruction tuning [1], which includes a pre-trained visual encoder, a pre-trained LLM, and a multimodal connector—a two-layer MLP—trained from scratch. Notably, the dataset size for LLaVA-1.5 (a general-purpose MLLM) is approximately 1 million instances. The intuition is that the multimodal connector, being an MLP with only a few million parameters, is the only component trained from scratch. Moreover, since the vision encoders are pre-trained with text supervision, it is easier to align it with LLM. Additionally, Geo-170K, a widely used geometry-focused multimodal training corpus proposed in G-LLaVA [2], comprises only about 9K unique images.
>
> In our experiments, we utilize a dataset of approximately 100K instances, as we specifically focus on task-oriented scenarios rather than general-purpose multimodal capabilities. To validate the adequacy of our dataset, we observed clear patterns of convergence in both the mixed-task results in Figure 3 and the training curves in Figure 4, indicating that our dataset size is sufficient for the targeted tasks. Furthermore, during the training of our Euclid model, we incorporated a dynamic dataset generation engine that advances the training stage only when the testing accuracy on intermediate evaluations reaches a predefined threshold of 0.99, which leads to robust convergence. Leveraging the flexibility of our dataset generation engine, we ensured that all designed tasks are fully learned by the model with sufficient training data.
>
> To further address your concern regarding dataset sufficiency, we conducted additional experiments by training the ConvNeXt-Large model on 1 million instances from a mixed-shape dataset for the PointLiesOnLine task (same setting as the last sub figure of Figure 3). We test the model on a separate test set (containing 1,500 instances) every 500 steps (i.e., 32K total training instances). The result is shown in Figure 19 of the updated paper, demonstrating that the model's test accuracy quickly approaches near 100% within fewer than 100K samples. This empirical evidence further confirms that the dataset volume used in our primary experiments is adequate for training the model effectively.
>
> > "Molmo predicts every potential answer, leading to their poor accuracy scores" – If this means, that instead of only answering the required/asked points on the line, Molmo outputs all the points on the line –– this seems less critical.
>
> Yes, this means Molmo frequently outputs all the points on the line instead of the required ones. Thanks for pointing this out and we will clarify this in the paper in our updated version.
>
> ### **Clarification Regarding Sampling Multiple Answers Per Question**
>
> Our evaluation revealed that MLLMs excel at identifying all letter names in a diagram (a global OCR task) but struggle to pinpoint the required points. Consequently, we assign a score of 0 for any response that includes at least one incorrect answer. To further mitigate this issue during evaluation, we append the original questions with detailed instructions (outlined in Appendix B). These instructions include specific answer templates for single-point (line) and multiple-point (line) responses. We observed that all evaluated models adhered to these instructions, indicating no difficulty in understanding verbal instructions and questions.
>
> Furthermore, following your suggestion, we incorporated two different evaluation metrics into the updated analysis:
>
> 1. Superset Accuracy: Accuracy is scored as 1 if the model's prediction is a superset of the required answer. The corresponding results are presented in Table 7 of the updated version of the paper.
> 2. Subset Accuracy: Accuracy is scored as 1 if the model's prediction is a subset of the required answer. These results are presented in Table 6 of the updated version of the paper.
>
>
> The overall performance trends remain consistent, with proprietary models outperforming open-source models. Notably, under the first evaluation setting (superset accuracy), Molmo-7B-D emerges as the best-performing open-source MLLM, and GPT-4o-mini achieves performance levels comparable to other proprietary models. This phenomenon aligns with our observation that GPT-4o-mini and Molmo-7B-D frequently enumerate all potential components of an answer. Consequently, their tendency to overgenerate leads to inflated scores under the superset evaluation metric, despite not accurately targeting the required answers.

---

> ### Author Response · Authors · 2024-11-25
> **Official Comment by Authors (2/2)**
>
> > How many solutions are sampled for each question? Is any form of nucleus sampling performed i.e. sampling multiple answers per question? Please provide details for the same.
>
> We use greedy decoding for each model to get deterministic results and a fair comparison. We only sample the answer once since it is deterministic.
>
> ### **Terminology of Shapes**
>
> The terminology ‘shape’ refers to a logical geometry shape (e.g., a triangle with a point connecting to the midpoint of the opposite edge, which is our first shape). During training, our geometry generation engine is able to generate many numerical instances of each logical shape (line 299-303). Specifically in this shape, the generation engine will first pick three random points from the canvas to form a triangle, and then connect the first point with the midpoint of the opposite edge. Then the generation engine will randomly pick 4 capital letters from the alphabet (A-Z). and assign the picked letters to 4 points on the diagram.
>
> > Following the above question, it might be important to increase the dataset by sampling more shapes from the geometric shape generation engine and then training on that dataset.
>
> Following our above response, if you mean numerical instances by ‘shapes’, our model is trained on near-infinite numerical instances of the three logical shapes. If you mean logical shapes by ‘shapes’, in our Euclid section, we do indeed train our model on a larger number of shapes and tasks. We completely agree that training on more logical shapes will be beneficial, but as noted in Section 4, making the training work on several human-curated logical shapes is already challenging, therefore we leave it to future works.
>
>
> ### **Adapting Negative-hard Mining Strategy for Automatic Curriculum**
>
> We appreciate the valuable suggestion. In our paper, we demonstrated the effectiveness of curriculum learning through a human-defined curriculum order. However, in scenarios involving a broader range of tasks and image types—such as developing an automatic geometry shape generation engine—it is crucial to adopt advanced strategies like negative-hard mining to automatically determine the difficulty of training instances. While such approaches are promising, incorporating more diverse tasks requires substantial design efforts and introduces additional complexities that fall beyond the scope of our current work. Therefore, we have added a discussion of this idea to the future work section of our paper (Line 515-520) and will explore and integrate such techniques in our future work.
>
>
> [1] Liu, H., Li, C., Li, Y., & Lee, Y. J. (2024). Improved baselines with visual instruction tuning. In Proceedings of the IEEE/CVF Conference on Computer Vision and Pattern Recognition (pp. 26296-26306).
>
> [2] Gao, J., Pi, R., Zhang, J., Ye, J., Zhong, W., Wang, Y., ... & Kong, L. (2023). G-llava: Solving geometric problem with multi-modal large language model. arXiv preprint arXiv:2312.11370.

---

> > ### Author Response · Authors · 2024-11-30
> >
> > Dear reviewer 2bjH:
> >
> > Thank you again for your insightful review! We hope our response has addressed your concerns satisfactorily. And we would be happy to engage further if you have any additional questions or suggestions.

---

### Author Response · Authors · 2024-11-25

## **General Response to All Reviewers**

We want to thank the reviewers for their constructive and thoughtful reviews! We feel encouraged that reviewers find our work’s topic to be fundamental, important (reviewer 2bjH) and novel (reviewer HWJq); our dataset to be valuable (reviewer 2bjH and hLZB); our empirically study to be extensive and interesting (reviewer E5Tw and hLZB); and our Euclid model to be strong and robust (reviewer 2bjH and E5Tw).

In response to reviewer comments, we have made a number of additions to our experiments and paper, which we summarize below:
- Added experiments studying the effect of scaling the size of large language models.
- Added experiments incorporating all seven tasks in Geoperception for Euclid’s training.
- Added an experiment expanding our training dataset to 1 million data instances, which shows that the dataset volume in our empirical study is sufficient.
- Additional evaluation of Cambrian-1 on Geoperception tasks, showing even MLLM trained on the same images and augmented text annotations still struggle in Geoperception.
- Additional evaluation of G-LLaVA on Geoperception tasks, showing the model struggles to properly follow instructions.
- Incorporating two additional evaluation metrics for Geoperception, showing the validity of our evaluation.

Again we thank reviewers and ACs for their great efforts and valuable insights! All updates will be consistently incorporated into the updated version of this paper.

---

> ### Comment · Reviewer_hLZB · 2024-11-25
>
> It is disappointing the authors waited until a day before the end of the discussion period to make their initial response. It is difficult in the remaining amount of time to review and discuss the large number of changes introduced. I will respond shortly today.
>
> I am also inclined to clarify that the impressions attributed to me above are not correct. I did not previously describe the evaluation as "insightful," but as "disappointingly shallow."

---

> > ### Author Response · Authors · 2024-11-26
> >
> > We apologize for the delay in responding, as we needed to train our models on a large newly-generated dataset and complete a few other suggested experiments. By ‘insightful’ we meant your positive review about the value of our diagnostic Geoperception benchmark — we are sorry for the oversight and we have revised the global response to clarify this.

---

### Meta-Review · Area_Chair_U7sm · 2024-12-23

**Metareview:**

This paper studies multimodal large language models (MLLMs) on the topic of low-level visual perception task. Specifically, the authors study low-level geometric understanding in MLLMs with a new benchmark and conduct empirical studies with proposed models. The manuscript was reviewed by four experts in the field. The recommendations are ("3: reject, not good enough", "5: marginally below the acceptance threshold", 2 x "6: marginally above the acceptance threshold"). The reviewers raised many concerns regarding the paper, e.g., unclear motivation and statement, unconvincing experimental benchmarks and evaluation results, etc. Considering the reviewers' concerns, we regret that the paper cannot be recommended for acceptance at this time. The authors are encouraged to consider the reviewers' comments when revising the paper for submission elsewhere.

**Additional Comments On Reviewer Discussion:**

Reviewers mainly hold concerns regarding unclear motivation and statement (Reviewer 2bjH, HWJq, hLZB), unconvincing experimental benchmarks and evaluation results (e.g., dataset size is quite small, utilize existing MLLM for data filtering, limited comprehensive geometric understanding, baselines selection)(Reviewer 2bjH, E5Tw, HWJq, hLZB). The authors' rebuttal could not fully address the above-listed concerns.

---

### Decision · Program_Chairs · 2025-01-22

Reject